# Active Learning of 3D Gaussian Splatting with Consistent Region Partition and Robust Pose Estimation

**Ruiqi Li, Yiu-ming Cheung** *
Department of Computer Science
Hong Kong Baptist University
`{csrqli,ymc}@comp.hkbu.edu.hk`

## Abstract

Radiance fields have been successful in reconstructing 3D assets for scenes presented in Virtual Reality and Augmented Reality (VR/AR). The general workflow of scanning objects with radiance field representation involves a heavy workload of capturing images depicting the object empirically by the user, and lacks feedback for the image collection stage. This would lead to potential repeated or deficient gathering of information, affecting the efficiency of the reconstruction workflow. In this paper, we therefore present an active learning algorithm for 3D Gaussian Splatting that guides the image capturing by estimating the pose of the most informative image. Specifically, our method first partitions the consistent regions in the model by analyzing the Gaussian attributes and visibility features. Then, we determine the informative region to explore by estimating the semantic feature variance of each Gaussian, which evaluates the quality of the Gaussian cloud from the semantic level features. Furthermore, we tackle the practical problem of noise in the pose of the collected image via a robust pose optimization method. Extensive experimental results on both synthetic and real-world scenes demonstrate the remarkable performance of our algorithm in active learning of the radiance field under both accurate and noisy pose conditions. [1]

## 1 Introduction

3D Gaussian Splatting (3DGS) Kerbl et al. (2023) has become popular in novel view synthesis of objects and scenes by learning a radiance field represented based on Gaussian ellipsoids. However, a major problem of 3DGS is that it requires a large number of images for training, which are desired to be placed densely to cover the scene. Capturing such images would require extensive and potentially repeated human labor. More importantly, timely feedback on the quality of the captured images is missing, leading to a potential oversampling of images in some object parts and undersampling in some others at the same time. This incurs the misalignment of the captured images and the required images for reconstructing the radiance field.

A practical solution to the above problem is the active learning of radiance field reconstruction, where image capturing and 3DGS model training are performed simultaneously in an online manner and provide mutual guidance for each other. Such an active reconstruction algorithm delivers the user with the next best poses for capturing images, where they can provide the model with the most information that is missing in the current model. Existing research on the active reconstruction mostly focused on evaluating the quality of rendered images to choose the next pose, and developed heuristic metrics such as predictive uncertainty or ray entropy Lee et al. (2022); Ran et al. (2023). However, these methods cannot directly generate new poses, and require randomly sampling some candidate poses first. Then, they evaluate the informativeness of the rendered images, which would ignore the occluded regions. Also, these metrics overlook the semantic features quality of object parts in the radiance field, which reflects the recognizability of object semantics and is an impor-

---

*Corresponding author is Yiu-ming Cheung (ymc@comp.hkbu.edu.hk).
[1]Code available at `https://github.com/csrqli/al-3dgs`.

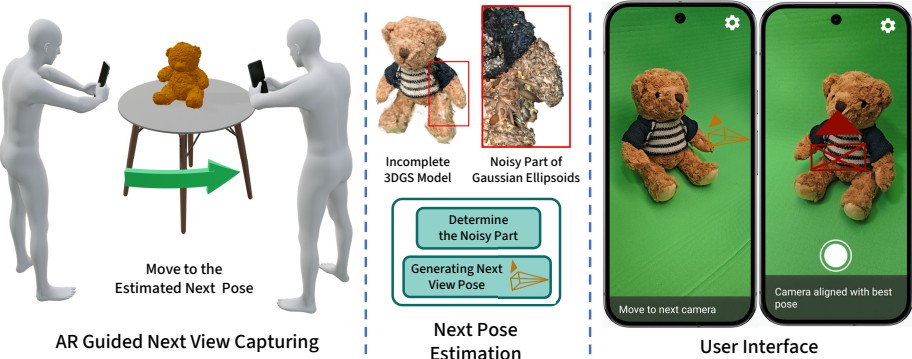

Figure 1: An application scenario of our active reconstruction algorithm. Our algorithm can be deployed on the server end, providing the next best pose in a front-end smartphone showing as virtual cameras to guide the image capturing.

tant indicator of the perceptual quality of the geometry and appearance of the reconstructed object. Moreover, previous works Jiang et al. (2023); Pan et al. (2022b) select the next views from a pre-collected dataset, assuming that the perfect camera pose is provided. However, from a practical perspective, the actual captured image might deviate from the desired best pose, leading to noise in the camera pose and poor reconstruction results.

In this paper, we develop a practical active reconstruction algorithm that solves the above challenges. Our method can directly analyze the noisy distribution of an incomplete 3DGS model under training, providing guidance for a progressive scanning of each object part, and finally composing a complete, semantically recognizable object with distinct appearance features.

Furthermore, to handle the noisy pose raised by the image capturing with, *e.g.*, a camera on a hand-held device, we design a pose optimization algorithm that can move the camera from the given next pose to the actual pose of collected image. Our algorithm runs in an online manner, providing the user with feedback on the noisy region of the current model and guidance to the next pose.

Conceptually, we apply a bottom-up strategy to capture the under-reconstructed object parts in sequence. Specifically, we propose an active reconstruction algorithm that can firstly divide the consistent part of the 3DGS models. These consistent parts share similar geometry, texture and visibility features. The divided consistent regions are captured by a set of adjacent cameras, where they can provide the best coverage for that region. We use the variance of the semantic feature of each Gaussian in the model to evaluate the quality of each Gaussian and determine the noise and uncertain region. Also, our method can adaptively decide the number of new poses in each image capturing step according to the size of the consistent region. Additionally, we employ a pose optimization with reprojection loss to optimize the actual pose of newly captured images, showing robustness under inaccurate poses.

Experiment results have demonstrated that the informativeness of the next best poses generated by our method outperforms those from the existing state-of-the-art method, evaluated quantitatively by the quality of synthesized novel views on benchmarks of the radiance field active reconstruction with perfect pose using a limited number of images. We also validated that our robust pose estimation can improve 3DGS model quality by reconstructing real-world objects under large pose noise. The contributions of our work are summarized as follows:

- We propose a bottom-up strategy for dividing and conquering the noisy consistent regions in reconstructing the radiance field of objects in an active manner.

- We propose to evaluate the noisy level of Gaussians with semantic feature variance to choose the next pose.

- We develop a pose fine-tuning method based on reprojection of visible Gaussian to provide depth supervision for optimizing the pose.

## 2 RELATED WORK

### 2.1 ACTIVE RECONSTRUCTION OF RADIANCE FIELD

Previous works investigated the active view selection problem of radiance fields with known actual poses of new images. Generally, they select the next view from a candidate set of images already collected with poses estimated. Among them, FisherRF Jiang et al. (2023) investigated the criterion for the next view, where they select the next image from a pre-collected set of images by maximizing the expected information gain. Earlier view selection methods build upon Neural Radiance Field (NeRF). NeurAR Ran et al. (2023) and ActiveNeRF Pan et al. (2022b) add uncertainty output into the neural network and optimize with Negative Log Likelihood (NLL) loss, which is then used to choose the next view. Lee et al. Lee et al. (2022) proposed to leverage the entropy of density distribution along the ray as the uncertainty metric for that pixel. Xue et al. Xue et al. (2024) proposed an uncertainty estimation method for NeRF based on a Bayesian neural network, and takes visibility into account in learning the uncertainty of rendered image. NeRF Director Xiao et al. (2024) proposed an evaluation framework for view selection. Other works Jin et al. (2023); Pan et al. (2023); Yan et al. (2023); Lee et al. (2023); Zhan et al. (2022) proposed different uncertainty or information gain estimation methods for rendered images of NeRF, while they cannot be directly used for 3DGS. Some previous works provided practical image capturing systems using cameras held by robot arms for the radiance field. ScanNeRF De Luigi et al. (2022) collects densely posed images from a pre-defined trajectory. Lee et al. Lee et al. (2022) used a wheeled robot and a five degrees of freedom manipulator to capture the estimated next best pose. However, they only consider the case when perfect actual poses are provided. Nevertheless, our work considers a more practical scenario, where the actual pose of the newly captured image deviates from the desired best pose, raised by error from the inaccurate pose of a camera held by a robot arm or a mobile device. Moreover, we utilize the explicit property of 3DGS, where we can generate directly informative poses without rendering and evaluating images. Thus, our method can run in an online manner and guide the users to image capturing.

Apart from object-level view selection, some other methods are proposed for the active reconstruction of large-scale scenes with neural implicit fields. These methods often take into account the path length when exploring the scene. Zeng et al. Zeng et al. (2022) proposed to reconstruct an implicit Truncated Signed Distance Function (TSDF) field, and leverage the neural network to approximate the information gain field in an implicit manner. Additionally, they deploy their method on a real drone. Active neural mapping and follow-up work Yan et al. (2023); Kuang et al. (2024) introduce an active exploration and mapping of indoor scenes with implicit representation. NARUTO Feng et al. (2024) defined an uncertainty metric used for the exploration of indoor scenes with the TSDF field. Additionally, some previous research Kwon et al. (2023); Chen et al. (2023); Adamkiewicz et al. (2022); Marza et al. (2023); Jiang et al. (2024); Chaplot et al. (2020); Jin et al. (2024); Li et al. (2024) leveraged the radiance field or implicit field and designed algorithms for tasks such as active object finding, navigation, and embodied agents, where they consider not only the maximization of the reconstruction quality but also navigation to the target and path efficiency. Our work can be extended to such a scene-level scenario by estimating the semantic feature variance metric for each object in the scene.

### 2.2 NEXT BEST VIEW PLANNING

Apart from the radiance field, some algorithms were proposed to solve the view planning or sensor planning problem in object-level scanning for data structures such as voxels, point clouds or building maps of the scene. The majority of these algorithms leverage robots equipped with depth sensors or use multi-view stereo methods to acquire accurate depth. Tarabanis et al. Tarabanis et al. (1995) designed a sensor planning algorithm for a robot vision system considering the detectability of feature points. GenNBV Chen et al. (2024) formulated the NBV problem as a Markov Decision Process (MDP), and learned a generalizable reinforcement learning model for next best view prediction using a simulator with hundreds of models. Yan et al. Yan et al. (2023) trained a sequential prediction model that can predict the best pose considering the previous input image and the model under reconstruction. They use reward functions such as the intersection of the reconstructed model with the ground truth to lead the training of best pose prediction.

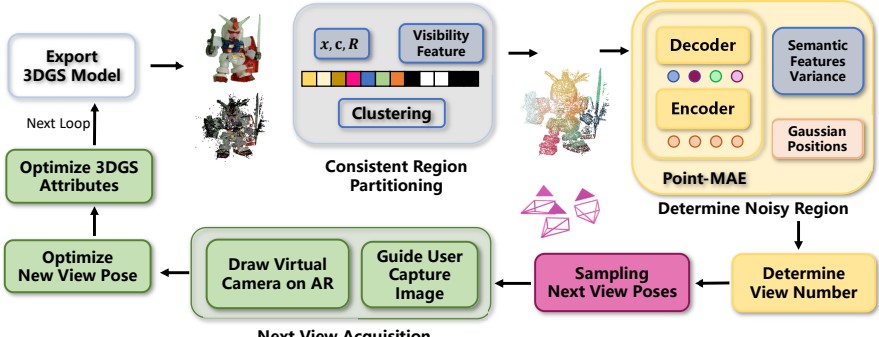

Figure 2: The above figure shows the pipeline of our active reconstruction algorithm. The next best pose estimation contains consistent region partitioning, noisy region determination and view sampling stages. The 3DGS model training, next pose estimation and guided image acquisition are performed iteratively.

Another line of works designed geometry-based methods Zeng et al. (2020); Scott et al. (2003); Lee et al. (2020); Scott et al. (2003); Mendoza et al. (2020); Daudelin & Campbell (2017); Kriegel et al. (2012; 2011) for view planning in object scanning with depth sensors, where accurate depth information is available. For example, Chen et al. Chen & Li (2005) established a uniform surface model by analyzing the surface curvatures of objects, which helps determine the pose that captures noisy parts. Mendoza et al. Mendoza et al. (2020) modeled the best pose prediction as a classification problem and built a 3D CNN that learns to predict the next best pose from a volume input. They trained on a dataset containing trajectories generated from 14 objects. SCVP Pan et al. (2022a) modeled the view planning as a set covering optimization problem. Different from the above works, our method is built upon the radiance field, which can generate dense reconstruction with photorealistic novel view renders with image input only.

Some other works Song et al. (2022); Bircher et al. (2016); Caccamo et al. (2016); Naazare et al. (2022); Sun et al. (2020); Wang et al. (2019) designed algorithms for drones and field robots to actively explore scenes. Yamauchi Yamauchi (1997) proposed the concept of frontiers for active exploration, which are locations lies on the intersection of explored and unknown areas in the scene. SCONE Guédon et al. (2022) defined the surface coverage metric and proposed a Monte Carlo integration over the volumetric representation to estimate the best pose. MACRONS Guédon et al. (2023) learned to predict a volume occupancy field and estimate the best view from previous observations in a self-supervised learning manner. Zhou et al. Zhou et al. (2020) built an exploration algorithm for drones that applies a hierarchical planning strategy. Recent scene-level active mapping approaches such as Naruto Feng et al. (2024), ActiveGAMER Chen et al. (2025a), and ActiveSGM Chen et al. (2025b) have further advanced 3DGS view planning, while our object-level method is a complementary direction targeting reconstruction in mobile capture settings.

Theoretically, the active mesh reconstruction problem is related to the art gallery problem O'Rourke (1987); Marzal, which investigates the minimum number of views to cover the surface of a known model. Some computational geometry algorithms Bonnet & Miltzow (2016) can solve the best view placement with known model geometry, whereas our algorithm reconstructs unknown models actively.

# 3 METHOD

## 3.1 ACTIVE LEARNING PIPELINE

In the original pipeline of 3DGS, a batch of $K$ images is first captured. Then, the Structure from Motion (SfM) algorithm is performed to estimate the camera pose of each image. After that, a 3DGS model $\mathcal{G}$ is trained by projecting the Gaussians into the image plane and performing rasterization to

get the rendered image:

$$\mathbf{c} = \sum_i \mathbf{c}_i \alpha_i G_i(\mathbf{x}) \prod_{j=1}^{i-1} (1 - \alpha_j G_j(\mathbf{x})), \tag{1}$$

where $i, j$ are the indices for Gaussians, $\mathbf{c}$ is the view-dependent color, determined by spherical harmonics, $\alpha$ is the density, and $G$ is a Gaussian function centered at the position $\mathbf{x}$: $G(\mathbf{x}) = \exp\left(-1/2 \cdot \mathbf{x}^\top \Sigma^{-1} \mathbf{x}\right)$, where the covariance matrix $\Sigma$ determines the rotation and scaling of Gaussian. During training, we minimize the difference between the image and the ground truth with a combination of $\mathcal{L}_1$ loss and structural similarity loss.

However, in our active reconstruction algorithm, the 3DGS model is learned in an online manner, where the image capturing and optimization take place in alternation. At the $k^{th}$ image capturing step, one or more images are captured from given poses:

$$\{\mathbf{P}_k^1, \mathbf{P}_k^2, ..., \mathbf{P}_k^l\} = A(\mathcal{G}^k), \tag{2}$$

where poses are chosen from the camera extrinsic space $\mathbf{P} \in SE(3)$. The acquisition function $A$ will analyze the model at the $k$ image capturing step $\mathcal{G}^k$ and estimate the next poses $\mathbf{P}$ while determining the number of poses needed to be captured $l$ adaptively.

The objective of the acquisition function $A$ is to provide the model with the most amount of information from images obtained at these poses, in order to minimize the effort of capturing images. In our work, we first build an algorithm that can find under-reconstructed parts of the 3DGS model. Specifically, our method first divides consistent regions of the current 3DGS model $\mathcal{G}^k$ and evaluates which regions are the most noisy and uncertain parts, thus requiring more images to cover. Then, our method can provide the informative camera pose in, for example, an Augmented Reality application to guide the user to collect images, as shown in Figure 1. These processes are repeated until the required number of images are collected.

The images collected via the human-guided operation will inevitably have poses that deviate slightly from the ideal estimated view poses. To address this pose error and ensure high-quality reconstruction, we utilize the estimated best poses as initializations for the actual image poses; then, we optimize the camera pose parameters of the newly collected images to ensure the pose accuracy of the new image used for reconstructing the 3D scene.

In practice, the actual pose of the images collected by the above guided operation performed by human users may deviate from the best poses. To solve this problem, we regard the estimated best poses as the initialization for actual pose, then, we optimize the pose of these new images together with the 3DGS model parameters.

## 3.2 INFORMATIVE VIEW ACQUISITION

In this section, we introduce our algorithm to determine the next poses for guiding the acquisition of new images. Generally, our method follows a bottom-up strategy, where the objects are scanned by parts individually. First, we will introduce the concept of local consistent regions and the algorithm to partition them. Then, we generate new view samples on local consistent regions that are not covered well by the data already collected. We determine these regions by estimating the variance of the per-Gaussian semantic feature, which is extracted by a self-supervised model trained on the large-scale dataset.

### 3.2.1 LOCAL CONSISTENT REGION DETERMINATION

The consistent regions are connected areas where the surface has smooth geometry, shares similar texture, and, most importantly, the complete region is often visible as a whole from a camera pose. We refer to the last property as the visibility feature, and will introduce an algorithm to determine it for 3DGS models. Once we determine the local consistent parts of the object, we employ a divide-and-conquer scheme to infer possible new observations added to improve the reconstruction performance of the most uncertain part. Figure 3 shows an example of the consistent region of a complex object composed of many parts.

We first estimate the visibility feature of the Gaussians. We refer to a region where the points share similar visibility features as a visibility-consistent region. An illustrative example is that a thin plate

has two sides; any camera that sees one side can see most of the surface on that side, and cannot see the other side. A visibility consistent region is mostly captured together. To determine the visibility feature, we first sampled a subset of $M$ 3D Gaussians, then reconstructed the surface of the model with Alpha Shape Edelsbrunner et al. (1983). After that, a set of $N$ camera positions are sampled uniformly on the surface of a hemisphere around the object. We estimate a visibility matrix $\Gamma$, where the $(m, n)$ entry of the matrix $\Gamma_{mn}$ is a binary indicator for the visibility of the $m^{th}$ point from the $n^{th}$ camera. A point is visible to the camera if it is not occluded by any mesh triangle, and we use the fragment buffer in mesh rasterization to determine visibility in the implementation. The $m$ row of the visibility matrix is the visibility feature for $m^{th}$ Gaussian. The visibility feature, concatenated with color and position constitutes the feature for each Gaussian:

$$\gamma = [\mathbf{x}/r, \mathbf{c}/\sqrt{3}, R, \Gamma_m/\sqrt{N}], \tag{3}$$

where $r$ is a predefined radius of the model, $R$ is the rotation of Gaussian, represented by quaternion. Each of the four components in feature vectors is normalized by its maximum value. Finally, we determine local consistent region by clustering the Gaussians with the above feature vector $\gamma$. We use the K-Means Burkardt (2009) algorithm to form clusters of Gaussians that form consistent regions.

Compared to previous work using image-based metrics for determining the next best poses, which is easily affected by image occlusion, our method determines the consistent region by visibility and is able to perform reconstruction without missing any informative region. Figure 3 shows that our method succeeds in distinguishing the two sides of the thin cymbal with different visibility.

### 3.2.2 SEMANTIC FEATURE VARIANCE SCORE

After partitioning the consistent regions, we have to determine which region is mostly noisy and under-reconstructed, and more images should be captured around it. Using deep features learned from large-scale datasets to evaluate the quality of data is a common practice in computer vision Zhang et al. (2018).

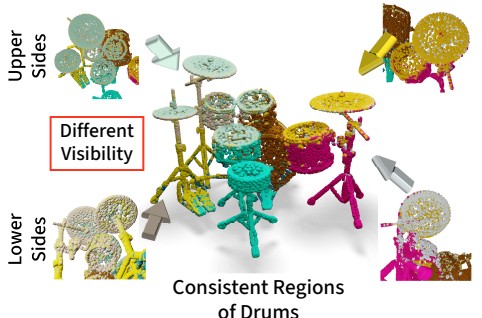

Figure 3: Our regions partitioning algorithm can divide Gaussians based on their visibility feature, which helps to find occluded surfaces and generate consistent next poses.

Therefore, we leverage the per-point semantic feature extracted from a point cloud analysis model to directly determine the noisy level of each Gaussian.

Specifically, we perform surface reconstruction with the current Gaussian model using Alpha Shape. We extract the vertices of the surface mesh as a point cloud. Then, we extract the per-point deep feature of the point cloud using a Point-MAE model Pang et al. (2022). This feature extraction model leverages an autoencoder with Transformer architecture, which is pretrained in a self-supervised manner by mask modeling and finetuning on the per-point segmentation task. We perform $T$ forward passes with different randomly sampled input points and dropout masks to get multiple instances of features. Then, we compute the norm of the average variance of the feature instances for each point to get the semantic variance score $S_{sem}$:

$$S_{sem} = ||\text{var}\{\frac{1}{T}\Sigma f_t(\mathbf{x})\}|| \tag{4}$$

where $f_t$ is a feature extractor used in $t^{th}$ forward pass with random dropout mask, $\mathbf{x}$ is one point. A larger variance score of the semantic feature instances implies that the semantic property of the point is unstable. Since the Transformer model captures sequential features, the region around that point has a higher noise level and poor quality Gal & Ghahramani (2016); Lakshminarayanan et al. (2017). Additionally, we use the following metric $S_{total}$ to choose the consistent region needed to be covered in the next image capturing step:

$$S_{total} = \lambda_1 \cdot \frac{1}{J} S_{sem}(\mathbf{x}_j) + \lambda_2 \cdot \min_k ||\hat{\mathbf{x}} - \tilde{\mathbf{x}}_k|| \tag{5}$$

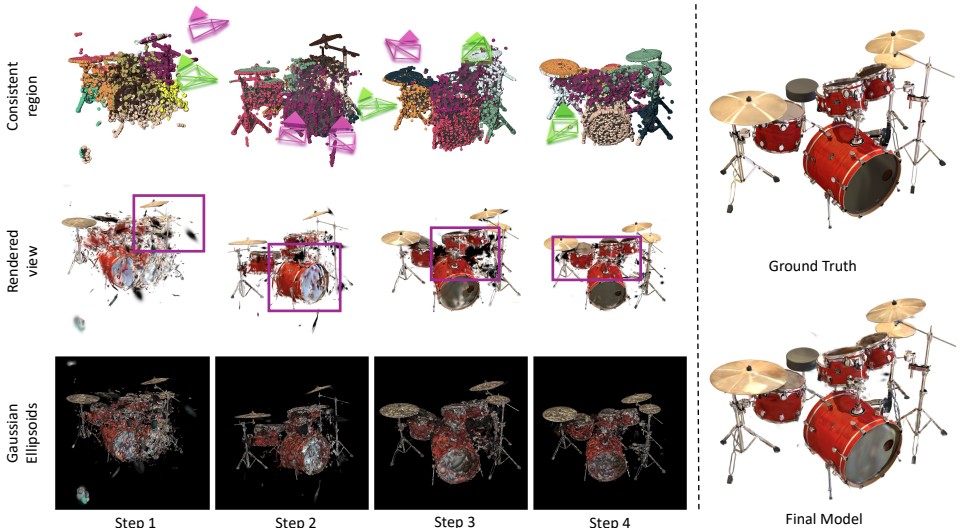

Figure 4: We visualize the process of the active reconstruction across multiple image capturing steps. The upper row shows the partitioned consistent region at each step, where the purple cluster is the noisy region chosen to be covered by the informative poses generated. The middle and lower rows show the under-reconstructed part in the 3DGS model. The model quality improves gradually by adding new views to cover the noisy regions.

where $\hat{\mathbf{x}} = \frac{1}{J}\Sigma_{j=1}^{J}\mathbf{x}_j$ is the center of points in the region, $\tilde{\mathbf{x}}_k$ is the center for chosen consistent region at $k^{th}$ image capturing step. The latter term implies that a larger distance with historical covered regions is encouraged in choosing the next region.

### 3.2.3 ADAPTIVE COVERAGE VIEW SAMPLE GENERATION

To generate the next best poses from a determined consistent region, we use the following sampling method. We choose the centering point in the consistent region and cast a line starting from that point along the normal direction. Then, we calculate the intersection point $p$ of that point and a bounding sphere with a given radius.

We sample camera position from a von Mises-Fisher distribution $v(\boldsymbol{\mu}, \kappa)$, where the $\boldsymbol{\mu}$ is the mean direction and $\kappa$ is the concentration parameter. We choose that intersection point on the bounding sphere as the mean direction $\boldsymbol{\mu} = p$, and use $\kappa = \eta_1 \cdot N_r/N_{all}$ as the concentration parameter, where $N_r$ is the number of points in the consistent region, $N_{all}$ is the number of total points. The number of poses sampled is also determined by this proportion of point number is $\eta_2 \cdot N_r/N_{all}$, where $\eta_1$ and $\eta_2$ are preset coefficients. After that, camera poses are built using the lookat method where the targeting point is the center consistent region. Since the von Mises-Fisher distribution is also known as the Gaussian distribution in the 3D direction, this design choice is to ensure that the sampled cameras are looking at the consistent region to cover. Our method can determine the number of views to capture and adaptively, according to the view needed for that consistent area, which further improves the efficiency of capturing the number of images needed.

As shown in Figure 4, our algorithm successfully divides the local consistent region of the cymbals in the drums, which is noisy in the current model. After we identify the most under-reconstructed consistent region using our method, we generate camera pose samples to cover that part. At the next image acquisition step, that part is well-reconstructed.

### 3.3 ACCURATE ACTIVE RECONSTRUCTION WITH ROBUST POSE ESTIMATION

After estimating the poses for covering the next consistent region, we use the desired pose to guide the user to capture the next poses, as shown in Figure 1. Once the image is collected, we optimize

Table 1: The quality of rendered images using active reconstruction under perfect pose.

| | 10 Views | | | 20 Views | | |
|---|---|---|---|---|---|---|
| **Method** | PSNR ↑ | SSIM ↑ | LPIPS ↓ | PSNR ↑ | SSIM ↑ | LPIPS ↓ |
| NeurAR Ran et al. (2023) | 18.94 | 0.774 | 0.241 | 23.71 | 0.819 | 0.172 |
| NeRF + Random | 16.430 | 0.809 | 0.254 | 22.614 | 0.890 | 0.111 |
| NeRF + Furthest | 19.24 | 0.735 | 0.227 | 26.00 | 0.812 | 0.144 |
| NeRF + ActiveNeRF Pan et al. (2022b) | 20.010 | 0.832 | 0.204 | 26.240 | 0.856 | 0.124 |
| NeRF + BayesRays Goli et al. (2024) | 14.950 | 0.798 | 0.282 | 22.504 | 0.887 | 0.115 |
| NVF Xue et al. (2024) | 18.691 | 0.821 | 0.228 | 21.736 | 0.925 | 0.162 |
| Plenoxel + FisherRF Fridovich-Keil et al. (2022); Jiang et al. (2023) | 20.670 | 0.824 | 0.205 | 24.513 | 0.876 | 0.157 |
| 3DGS + Random | 22.493 | 0.873 | 0.112 | 28.732 | 0.939 | 0.053 |
| 3DGS + Furthest | 18.782 | 0.764 | 0.200 | 22.346 | 0.814 | 0.143 |
| 3DGS + ActiveNeRF Pan et al. (2022b) | 22.979 | 0.876 | 0.111 | 26.610 | 0.905 | 0.081 |
| 3DGS + FisherRF Jiang et al. (2023) | 23.681 | 0.883 | 0.102 | 29.525 | **0.944** | 0.043 |
| Ours | **25.542** | **0.896** | **0.063** | **30.186** | 0.943 | **0.033** |

the 3DGS model using the new image. In practice, the actual poses of the captured image might differ from the given best view poses, considering that error is caused by the position of the handheld device and the actual desired pose. Thus, we further optimize the actual pose of the captured images.

Specifically, we first freeze the 3DGS attributes and optimize only the relative pose between the actual pose of the new image and the desired best pose:

$$T^* = \arg\min_T \mathcal{L}_{pose}\left(T\mathbf{P}, \mathbf{P}\right). \tag{6}$$

We would like to optimize the relative rigid transformation $T \in SE(3)$ between the estimated best pose $\mathbf{P}$ and the actual new pose $T\mathbf{P}$. Numerically, we optimize the Lie algebra $\tau = \text{Log}\,T$ to avoid an ill transformation matrix. We optimize the relative pose with the following objective function:

$$\mathcal{L}_{pose} = \mathcal{L}_{rgb}\left(\mathbf{c}(T\mathbf{P}), \hat{c}(\mathbf{P})\right). \tag{7}$$

We use the photometric loss here to minimize the distance between the image rendered by the model $\mathbf{c}(T\mathbf{P})$ and the acquired new ground truth image $\hat{\mathbf{c}}(\mathbf{P})$. During pose optimization, we freeze the learning of Gaussian attributes, which can avoid the infection of deviated pose to the Gaussian model. After the relative pose optimization, we continue the learning of the 3DGS model parameters. An overview of our pipeline is shown in Figure 2.

## 4 EXPERIMENTS

### 4.1 BACKGROUND

**Implementation Details.** We use 3D Gaussian Splatting as the reconstruction algorithm. Our active reconstruction algorithm follows the same optimizer and learning rate for each attribute of the 3DGS model. The image acquisition is performed every 300 steps, after which the model is reinitialized as random points. To cluster Gaussians and perform consistent region partition, we sample a subset of 10K Gaussians from the model, and use the K-Means algorithm and a relative tolerance of $1e^{-4}$. After all images are collected, the model is further trained for $10,000$ steps.

**Baselines.** We chose the following active reconstruction methods as a comparison with our method: **1.** NeurAR chooses the most uncertain next view evaluated by the additional variance output. **2.** ActiveNeRF, which acquires the image with the most posterior uncertainty. **3.** FisherRF, which uses Laplace's approximation to estimate the expected information gain. **4.** BayesRay quantifies the uncertainty of radiance field with Laplacian approximation, and the most uncertain image is chosen. **5.** Random denotes randomly selecting views, and Furthest denotes selecting the most distant view from existing ones. We choose NeRF, Plenoxels Fridovich-Keil et al. (2022) and 3DGS as reconstruction backbones.

### 4.2 ACTIVE LEARNING WITH PERFECT POSE

We compare the reconstruction performance using 10 or 20 total views for all eight objects in the NeRF-Synthetic dataset Mildenhall et al. (2020), and report the averaged image quality. For our

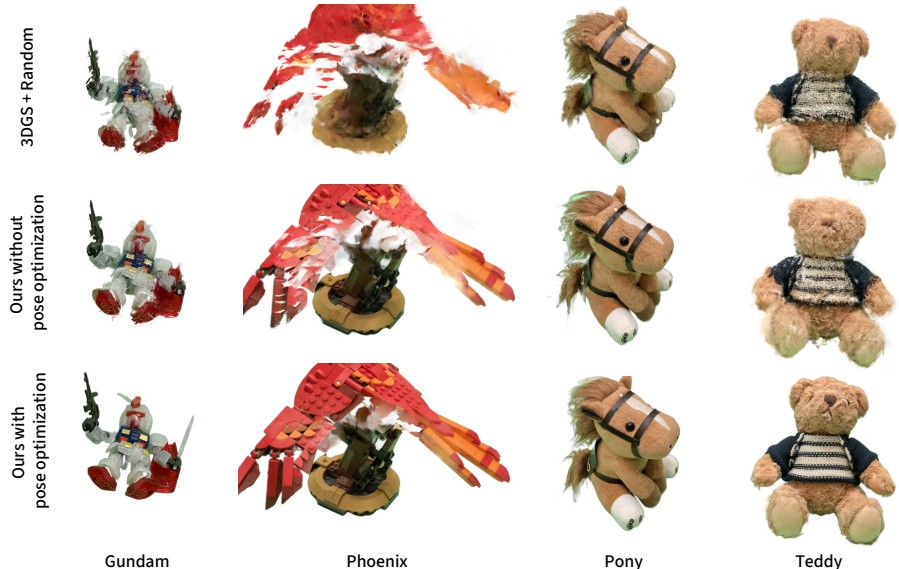

Figure 5: The qualitative results of our active reconstruction with a handheld device. We compare the visual results of rendered novel views.

method, we first capture three initial images uniformly as initialization. Then, the best poses are estimated and corresponding images are rendered in Blender and used for training. From the results shown in Table 1, our method surpassed largely the performance of previous methods, such as ActiveNeRF and FisehrRF, in all quantitative metrics. In terms of the PSNR metric, our method outperforms FisherRF by 1.9.

The results of using 20 total view settings are shown in Table 1. Our method is superior to the previous state-of-the-art method, FisherRF, in both PSNR and LPIPS and shows comparable results in SSIM. We outperform FisherRF by 0.6 in terms of the PSNR metric. However, the performance increase is not as significant as the 10 views. A plausible reason is that as the total number of images increases, the redundancy of information in the given views also increases. We visualize the process of next view generation and consistent region partition of the `drums` scene in Figure 4.

## 4.3 ACTIVE RECONSTRUCTION WITH NOISY POSE

We reconstruct four new real-world objects using our active reconstruction system, and compare the visual quality of the reconstructions with and without pose optimization. We collect the next image by manually matching the next best pose with the pose of the smartphone camera using the view synthesis results from the current 3DGS model. Therefore, the pose of the collected image roughly aligns with the

Table 2: Ablation study results of our method.

|  | PSNR | SSIM | LPIPS |
|---|---|---|---|
| w/o $\Gamma$ | 21.271 | 0.863 | 0.137 |
| w/o Distance | 22.438 | 0.866 | 0.090 |
| w/o Point-MAE | 24.584 | 0.877 | 0.074 |
| Full | **25.542** | **0.896** | **0.063** |

desired pose, with some noise on the camera pose raised by manually capturing left for our algorithm to deal with. The qualitative results of active reconstruction with noisy pose are shown in Figure 5. Without fine-tuning the actual pose of new captures, the noise in pose lowers the visual quality of the model. For example, the wings of the phoenix toy fail to reconstruct and become a white background. As a comparison, our robust pose optimization technique significantly improves the quality of the overall reconstruction.

## 4.4 ABLATION STUDY

We compare the following variants of our method and baseline on 8 scenes from Blender dataset using 10 total images: **1.** Remove the visibility feature $\Gamma$ in Equation 3 when clustering local region; **2.** Remove the larger distance term in Equation 5 when choosing the most uncertain region; **3.** Replace Point-MAE feature with Fisher Information implemented in FisherRF to validate the effectiveness of semantic feature variance in estimating uncertainty. The results of view synthesis are shown in Table 2, and removing any component would lead to a performance decrease, which validates the effectiveness of each individual component.

## 5 CONCLUSION

In this paper, we have proposed a practical system for online image acquisition and 3DGS reconstruction. Our method applies a divide-and-conquer strategy, which captures the local consistent regions adaptively to ensure complete coverage of the object. Experiment results have demonstrated the superior performance of our method. Future work can be made to improve the efficiency of our method or adapt it to large-scale environments.

## 6 ACKNOWLEDGEMENT

This work was supported by the Guangdong and Hong Kong Universities "1+1+1" Cross-Campus Research Collaboration Scheme with the grant: 2025A0505000004.

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

# A  IMPLEMENTATION DETAILS

## A.1  VISIBILITY FEATURE EXTRACTION

Firstly, we perform Alpha Shape Edelsbrunner et al. (1983) with alpha value equals 0.05 to reconstruct the triangle surface of the object from points in the exported Gaussians. The attributes of the original Gaussian are assigned to each vertex in the reconstructed mesh by mapping each vertex with the attributes of the closest Gaussian.

We sample $N = 100$ cameras randomly located on the surface of a sphere enclosing the object and looking at the object center to determine the visibility matrix $\Gamma$. To perform visibility checking for each vertex in the mesh, we leverage the output of the Pytorch3d rasterizer Ravi et al. (2020). We first identify which faces are rendered at each pixel. Then, we use the depth information stored in the depth buffer to find the closest face to the viewer. Specifically, for the $n^{th}$ camera, we map each pixel to the vertex that appears in the pixel. We iterate through the rendered pixel, marking the faces projected to each pixel with the minimum depth. Then, we find the vertex that belongs to the faces, which are regarded as the visible vertex for the $n^{th}$ camera. We assign binary values indicating visible or not for each vertex mask in the $n^{th}$ column of the visibility matrix $\Gamma$. Finally, each $m^{th}$ row of $\Gamma$ forms the visibility feature for that Gaussian point.

We compare quantitatively the image quality by the following three metrics: Peak Signal to Noise Ratio (PSNR), Structural Similarity (SSIM) and Learned Perceptual image Patch Similarity (LPIPS) with `alexnet`.

## A.2  DEEP SEMANTIC FEATURE

We extract deep semantic features used in clustering the consistent regions for each Gaussian point. We use a Point-MAE model Pang et al. (2022), where the encoder has 12 Transformer blocks. The model is pre-trained in a masked auto-encoding manner on the ShapeNet dataset Chang et al. (2015), and then finetuned on the ShapeNetPart dataset for the segmentation task. We use the feature extracted by the transformer encoder, which has 512 dimensions for each point.

To obtain the per-point semantic feature, we perform a total of $T = 10$ times forward pass. In each inference, we randomly sample 90 percent of the points as input, and dropout is performed for the model parameters with a probability of 0.8 with different draws. Finally, we compute the norm of the variance of features to get the semantic variance score.

Finally, we use K-Means clustering Burkardt (2009) with the combined and normalized features to find the clusters of points within the same consistent region. We use K-Means with a number of clusters equals 5, the maximum number of iterations equals 300, and the relative tolerance of the difference in the cluster centers equals $1^{-4}$.

## A.3  VIEW SAMPLING

In our active reconstruction with perfect poses on the Blender dataset experiments, we use a concentration parameter $\eta_1 = 0.1$, and the coefficient for determining the number of generated cameras $\eta_2 = 1$. In real-world scenarios in AR-based active reconstruction, users can adjust these parameters to fit the size and geometry complexity of the object.

## A.4  3DGS TRAINING

For selecting the best poses during active reconstruction, we train the 3DGS using the images already collected from a random initialization every 500 steps. Finally, the model is trained for 10K steps. For blender scenes, we use a densification interval of 100 steps. We reset the opacity every 3000 steps. The densification operation is always performed during active view selection, but is stopped after the $7000^{th}$ step of the final model training.

# B    ALGORITHM PSEUDOCODE

---

**Algorithm 1** Active Next Best View Selection

---

1: **Input:** Current 3DGS model $G_k$, acquisition history $\{P^1, ..., P^{k-1}\}$
2: **Output:** Next view poses $\{P_k^1, ..., P_k^l\}$
3: Sample $M$ Gaussians $\{g_i\}_{i=1}^M$ from $G_k$
4: Sample $N$ camera views $\{v_j\}_{j=1}^N$ uniformly around object
5: Initialize visibility matrix $\Gamma \in \{0, 1\}^{M \times N}$
6: **for** each view $v_j$ **do**
7:     Render alpha shape mesh from $G_k$
8:     **for** each Gaussian $g_i$ **do**
9:         Project $g_i$ into view $v_j$
10:         **if** $g_i$ is visible in $v_j$ (not occluded via rasterization) **then**
11:             $\Gamma[i, j] \leftarrow 1$
12:         **else**
13:             $\Gamma[i, j] \leftarrow 0$
14:         **end if**
15:     **end for**
16: **end for**
17: **for** each Gaussian $g_i$ **do**
18:     Construct feature vector $\gamma_i = [x_i/r, c_i/\sqrt{3}, R_i, \Gamma_i/\sqrt{N}]$
19: **end for**
20: Cluster $\{\gamma_i\}$ into consistent regions $\{R_j\}_{j=1}^C$
21: **for** each region $R_j$ **do**
22:     Extract surface mesh and sample points $\{x_j\}$
23:     **for** $t = 1$ to $T$ **do**
24:         Compute deep features $f_t(x_j)$ using Point-MAE with dropout
25:     **end for**
26:     Compute semantic variance $S_{\text{sem}}(x_j) = \|\text{var}_t f_t(x_j)\|$
27:     Compute region score:
28:     $S_{\text{total}}^j = \lambda_1 \cdot \frac{1}{|R_j|} \sum S_{\text{sem}}(x_j) + \lambda_2 \cdot \text{dist}(R_j, \text{history})$
29: **end for**
30: Select region $R^* = \arg\max_j S_{\text{total}}^j$
31: Determine center and surface normal of $R^*$
32: Sample view directions from von Mises-Fisher distribution centered at bounding sphere intersection
33: Generate view poses $\{P_k^1, ..., P_k^l\}$ using look-at orientation toward $R^*$ center
34: **return** $\{P_k^1, ..., P_k^l\}$

---

To facilitate reproducibility and clarify the operational details of our active view selection strategy, we provide a pseudocode description of our method in Algorithm 1. This algorithm outlines the full pipeline for estimating the next best camera poses in our bottom-up active reconstruction framework.

# C    ADDITIONAL ABLATION STUDY

## C.1    INVESTIGATING THE ROBUSTNESS OF CLUSTERING PARAMETERS

In separating local consistent regions, the clustering parameter is of great significance in the determination of parts of the object for generating the informative views. Intuitively, each cluster should contain a part that is independent of other object parts. Specifically, we investigate the influence of the number of clusters on the overall reconstruction performance. We test with a total of 10 views on all 8 scenes from the Blender dataset, setting the number of clusters from $\{3, 5, 7\}$. In practice, the number of clusters is given by the user based on their observations of the geometrical complexity of the object to reconstruct. The results are shown in Table 3.

Table 3: Ablation study on the number of clusters in consistent region partition.

| Number of Cluster | PSNR ↑ | SSIM ↑ | LPIPS ↓ |
|---|---|---|---|
| 3 | 24.913 | 0.889 | 0.074 |
| 5 (Default) | **25.542** | **0.896** | **0.063** |
| 7 | 24.992 | 0.873 | 0.079 |

Table 4: Ablation study on the number of semantic feature samples.

| Number of Samples | PSNR ↑ | SSIM ↑ | LPIPS ↓ |
|---|---|---|---|
| 5 | 25.178 | 0.884 | 0.077 |
| 10 (Default) | **25.542** | **0.896** | **0.063** |
| 15 | 25.976 | 0.901 | 0.061 |

Table 5: Ablation study on the number of samples in the visibility feature calculation in consistent region partition.

| Number of Sampled Camera $N$ | PSNR ↑ | SSIM ↑ | LPIPS ↓ |
|---|---|---|---|
| 50 | 24.072 | 0.851 | 0.080 |
| 100 (Default) | **25.542** | **0.896** | 0.063 |
| 200 | 25.497 | 0.895 | **0.061** |

The results demonstrate that the performance of our method varies slightly when using different numbers of clusters. This indicates that our method shows robustness to the setting of hyperparameters, demonstrating little performance variance against the variations of parameters.

## C.2 Investigating the Number of Semantic Feature Variance Samples

In this ablation study, we explore the effect of different sample sizes $\{5, 10, 15\}$ on estimating Semantic Feature Variance using the Point-MAE model. Our goal was to determine the optimal number of samples needed for accurate variance estimation and informative view estimation without excessive computational cost. We reconstruct all 8 objects from the Blender dataset with 10 total views. The results showed that using only 10 samples led to a balance between variance estimation and reconstruction performance.

As shown in Table 4, 5 samples are insufficient to capture the Semantic Feature Variance that exists in the noisy 3DGS model. Conversely, increasing the sample size to 15 did not significantly improve the accuracy of variance estimation compared to using 10 samples. The computational cost grows linearly with the number of samples. Therefore, using 10 samples is both practical and efficient, optimizing the active learning performance without unnecessary inference of the model.

## C.3 Investigating the Samples in Estimating the Visibility Matrix

In this ablation study, we investigate the sampled view in visibility matrix estimation. Specifically, we test different numbers of sampled cameras $N$ that are used to calculate the visibility feature. We report the results from all objects in the Blender dataset, reconstructed with $N \in 50, 100, 200$. The computational cost of visibility determination grows linearly with the number of sampled cameras $N$. The quality of the visibility matrix would influence the determination of the local consistent region, therefore further influencing the performance of reconstruction. Our results are shown in Table 5.

The quantitative results show that our visibility feature estimation requires 100 sampled cameras to evaluate the visibility of the objects from all possible viewing angles, conveying the features of occlusion between different parts to the local consistent region partitioning algorithm. Using only 50 samples can lead to a performance degradation, and using 200 samples increases the cost with no significant gain.

Table 6: The per-scene quantitative results of rendered images using active reconstruction under perfect pose with 10 total views.

| Scene | PSNR ↑ | SSIM ↑ | LPIPS ↓ |
|---|---|---|---|
| chair | 24.712 | 0.917 | 0.049 |
| drums | 21.840 | 0.895 | 0.059 |
| ficus | 27.934 | 0.946 | 0.028 |
| hotdog | 28.642 | 0.937 | 0.049 |
| lego | 23.230 | 0.859 | 0.079 |
| materials | 24.196 | 0.882 | 0.065 |
| mic | 29.304 | 0.965 | 0.023 |
| ship | 24.475 | 0.767 | 0.151 |
| Average | **25.542** | **0.896** | **0.063** |

Table 7: The per-scene quantitative results of rendered images using active reconstruction under perfect pose with 20 total views.

| Scene | PSNR ↑ | SSIM ↑ | LPIPS ↓ |
|---|---|---|---|
| chair | 31.378 | 0.966 | 0.016 |
| drums | 24.155 | 0.931 | 0.040 |
| ficus | 33.642 | 0.980 | 0.010 |
| hotdog | 33.116 | 0.965 | 0.027 |
| lego | 29.028 | 0.933 | 0.033 |
| materials | 28.808 | 0.941 | 0.026 |
| mic | 33.292 | 0.982 | 0.012 |
| ship | 28.072 | 0.850 | 0.096 |
| Average | **30.186** | **0.943** | **0.033** |

# D  JUSTIFICATION FOR MAJOR DESIGN CHOICE

## D.1  BOTTOM-UP STRATEGY VS RENDERED IMAGE BASED METHODS

Our method introduces a novel bottom-up strategy that directly generates informative camera poses, making it the first approach capable of estimating the next view pose without requiring candidate sampling. This design choice marks a significant departure from prior methods Jiang et al. (2023); Pan et al. (2022b); Ran et al. (2023) that rely on rendering from a pre-defined set of sampled views and then evaluating them using uncertainty or information gain metrics. Those approaches suffer from inefficiencies and often miss occluded or structurally ambiguous regions. In contrast, our method leverages the structure of the 3DGS model to identify under-reconstructed parts and generate poses that target them precisely. This strategy not only streamlines the active view selection pipeline but also enables real-time guidance during image acquisition, which is crucial in practical scenarios involving handheld or mobile devices.

Additionally, traditional active view selection often relies on sampling a set of candidate views, rendering images from these viewpoints, and scoring them using informativeness heuristics such as entropy or uncertainty. However, this process is both computationally expensive and vulnerable to occlusion artifacts, often overlooking poorly reconstructed or unseen regions. In contrast, our bottom-up strategy directly analyzes the geometric and visibility characteristics of the model to identify areas to capture. By estimating visibility-consistent regions and their semantic uncertainty, we can generate next-best views without requiring rendering or post-hoc evaluation. This direct approach not only accelerates the acquisition process but also increases coverage efficiency and robustness in real-world scenarios where perfect pose alignment cannot be guaranteed.

Table 8: The per-scene quantitative results of AR-based active reconstruction with pose optimization.

| Scene | PSNR ↑ | SSIM ↑ | LPIPS ↓ |
|---|---|---|---|
| gundam | 30.363 | 0.963 | 0.029 |
| phoenix | 19.206 | 0.873 | 0.152 |
| hourse | 26.959 | 0.903 | 0.077 |
| teddy | 28.438 | 0.898 | 0.073 |
| Average | 26.241 | 0.909 | 0.083 |

Table 9: The per-scene quantitative results of AR-based active reconstruction without pose optimization.

| Scene | PSNR ↑ | SSIM ↑ | LPIPS ↓ |
|---|---|---|---|
| gundam | 24.962 | 0.878 | 0.080 |
| phoenix | 17.755 | 0.855 | 0.199 |
| hourse | 20.871 | 0.743 | 0.174 |
| teddy | 24.256 | 0.846 | 0.136 |
| Average | 21.961 | 0.831 | 0.147 |

## D.2 SEMANTIC FEATURE VARIANCE

To identify noisy or incomplete regions in the 3DGS model, we propose a semantic feature variance metric computed per Gaussian point. This metric, derived from multiple stochastic passes through a pretrained Point-MAE encoder, quantifies the uncertainty of semantic representation and reflects the overall distortion of each Gaussian. Methodologically, our approach build upon the technique proposed by Kendall & Gal (2017) for estimating epistemic uncertainty. They utilize Monte Carlo dropout to approximate the posterior distribution of model weights and estimate uncertainty via the variance of the model's predictive distribution across multiple stochastic forward passes. Similarly, we implement stochastic dropout within the Transformer blocks of our Point-MAE encoder. However, instead of measuring variance in the final output, we compute the variance in the semantic feature space. This adaptation is crucial for our reconstruction task: high variance in this feature embedding indicates that the model is uncertain about that region, and thus requires further visual coverage. Therefore, targeting regions with high semantic feature variance is a principled strategy to improve the model. In spirit, this is analogous to using LPIPS Zhang et al. (2018) as a perceptual metric to evaluate image quality, but applied at the semantic feature level in 3D space. This approach offers significant advantages over simpler 3DGS-native cues, such as RGB/opacity variance and Fisher Information, which are local and low-level indicators. Unlike these heuristics, our metric allows us to directly assess the high-level semantic quality and informativeness of the 3D Gaussian cloud representation, providing a principled, robust, and geometry-aware way to determine which parts of the model require further refinement.

## E   ADDITIONAL RESULTS

### E.1   RUNNING TIME AND STANDARD DEVIATION OF RESULTS

The running time for local consistent region partition, calculating the variance of semantic feature and view generation is respectively 4.13, 3.7 and 0.01 seconds. All the view synthesis results of our method reported in the paper are based on 3 runs. In both tables, the Standard Deviation across runs is less than 0.1 for PSNR and 0.01 for SSIM and LPIPS.

Table 10: The per-scene quantitative results on the Objaverse dataset.

| Method | FisherRF | | | Ours | | |
|--------|----------|--------|----------|---------|--------|----------|
| Scene | PSNR ↑ | SSIM ↑ | LPIPS ↓ | PSNR ↑ | SSIM ↑ | LPIPS ↓ |
| statue | 38.365 | 0.985 | 0.008 | 42.929 | 0.993 | 0.004 |
| warrior | 33.851 | 0.962 | 0.021 | 28.046 | 0.912 | 0.056 |
| bucket | 23.372 | 0.856 | 0.085 | 39.639 | 0.983 | 0.012 |
| rose | 37.960 | 0.986 | 0.007 | 43.752 | 0.994 | 0.003 |
| lion | 25.472 | 0.860 | 0.081 | 31.303 | 0.923 | 0.054 |
| bacanal | 27.928 | 0.940 | 0.028 | 36.161 | 0.983 | 0.008 |
| rhino | 37.969 | 0.972 | 0.023 | 43.423 | 0.987 | 0.013 |
| turtle | 36.316 | 0.971 | 0.016 | 41.318 | 0.987 | 0.007 |
| **Average** | 32.654 | 0.942 | 0.034 | **38.321** | **0.970** | **0.020** |

### E.2 PER-SCENE RESULTS ON BLENDER DATASET

We provide the per-scene quantitative results for the active reconstruction with perfect pose experiments on 8 scenes from the Blender dataset. The results are shown in Table 6 and Table 7. The chair scene gains the most significant performance improvements by increasing the view number from 10 to 20, with the PSNR metric increasing by 6.67. We also provide the visualization results of rendered novel views in Figure 6 and Figure 7.

### E.3 QUANTITATIVE RESULTS OF ACTIVE RECONSTRUCTION WITH NOISY POSE

In active reconstruction with 4 real-world scenes with noisy pose, we acquire a total of $30 \sim 40$ images for each scene, according to their reconstructed model quality. We chose every 8 images as the test set, and provide the quantitative results in the four objects captured by our active reconstruction application, in Table 8 and Table 9. From the average results, we can see that our pose optimization increases the performance of rendered novel views largely. We provide the training video of the active reconstruction of gundam scene in the supplementary files.

### E.4 RESULTS ON THE OBJAVERSE DATASETS

Apart from the standard object-level radiance field reconstruction dataset, the Blender dataset, we further validate the effectiveness and generalization ability of our method on more objects from a high-quality dataset of 3D Models scanning from real-world objects, the Objaverse dataset Deitke et al. (2023). Specifically, we select and reconstruct 8 objects from the Objaverse dataset, containing objects like statue, bucket, and so on, with variance in geometry and scale. Regardless of different object classes, we perform active learning for the Objaverse dataset with the same hyperparameters as the Blender datasets for both methods. We compare the results of our method and the FisherRF Jiang et al. (2023) both quantitatively and qualitatively. In Table 10, we compare the quality of novel views reconstructed by both methods. Our method outperforms FisherRF in image quality metrics for most objects. For the averaged results, our method is superior to FisherRF, with an increase in PSNR of 5.7. This indicates that the superior performance of our method is generalizable to variant objects. We also visualize the reconstructed novel views of both methods, together with the ground truth in Figure 8 and Figure 9.

### E.5 ESTIMATED NEXT BEST VIEWS DISTRIBUTION VISUALIZATION

To understand the feature of the next views generated by our method, we visualize the distribution of all views selected to reconstruct the object, in the 10 total views case, together with the object to reconstruct. The distribution of views is shown in Figure 10 and Figure 11. For each object, our estimated view is distributed evenly around the object, capturing the object from an informative

viewing pose. The quality of generated views is stable across different types of objects, showing robustness against the shape appearance of different objects.

## F    LIMITATIONS AND FUTURE WORKS

Our method is designed for object-level active 3DGS reconstruction but has several limitations. It inherits weaknesses from the 3D Gaussian Splatting (3DGS) backbone, particularly under transparent or reflective surfaces where geometry is poorly captured. While our pose refinement improves robustness, the system requires a clear captured image whose pose is near the estimated next view without significant motion blur. Additionally, the current design targets single-object scenes and relies on fixed semantic encoders, which may limit generalization in cluttered or open environments.

In the future, we could extend our method to scene-level reconstruction by incorporating semantic segmentation and multi-object reasoning. We could also improve the robustness of our pose estimation in dealing with the large initial drift by integrating learning-based estimators, such as DUSt3R and its variants Wang et al. (2025); Leroy et al. (2024); Wang et al. (2024); Yang et al. (2024). Moreover, adapting the semantic encoder with other point cloud understanding networks Zhao et al. (2021); Park et al. (2022) or introducing class-aware view selection policies could help generalize across object categories or specific downstream tasks.

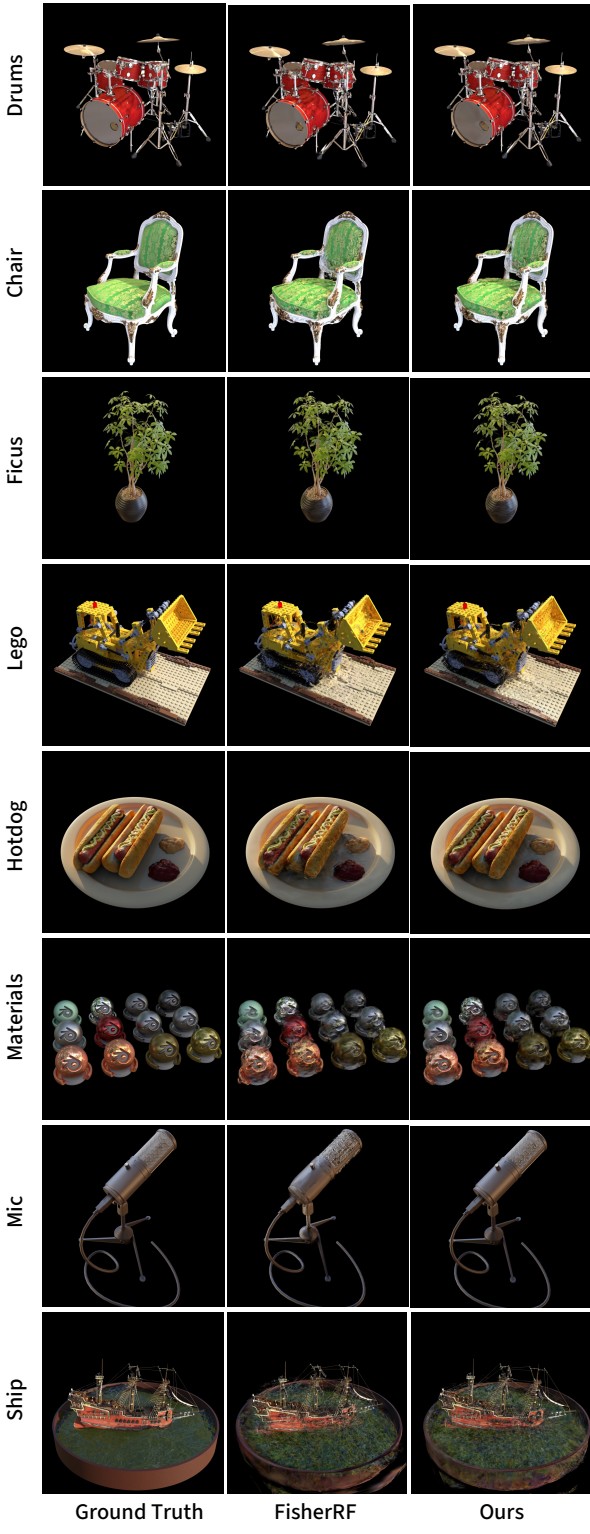

Figure 6: The per-scene qualitative results of rendered images using active reconstruction under perfect pose with 10 total views.

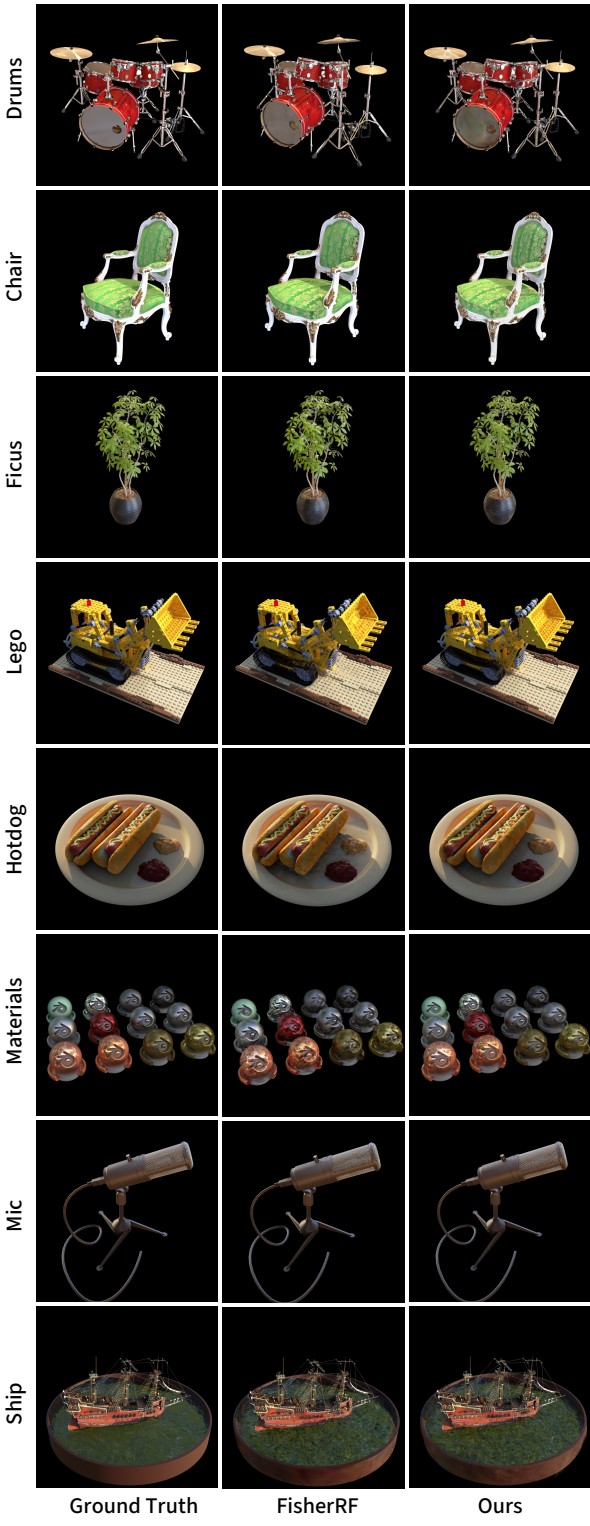

Figure 7: The per-scene qualitative results of rendered images using active reconstruction under perfect pose with 20 total views.

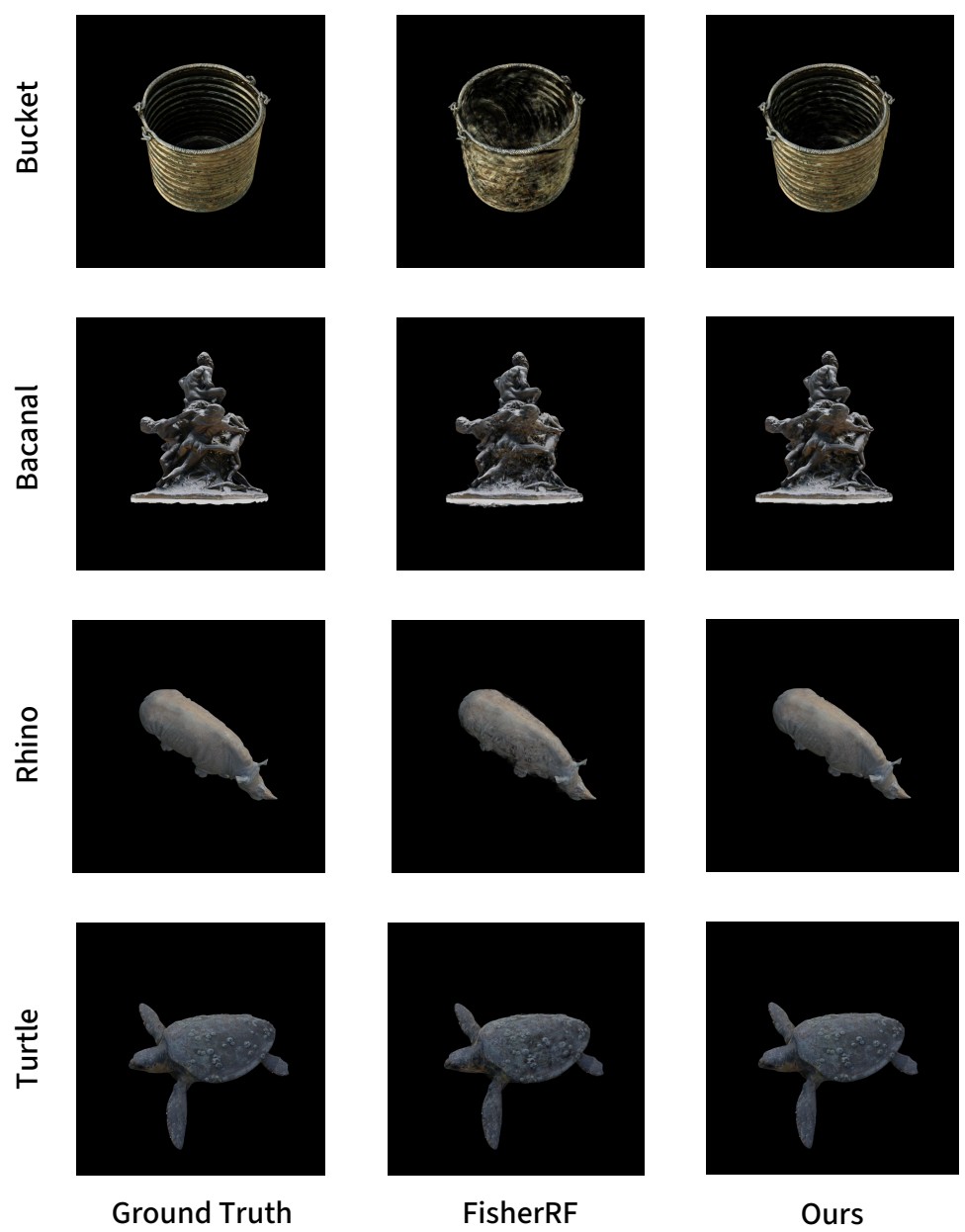

Figure 8: The per-scene qualitative results of rendered images using active reconstruction using 20 total views on the scenes from the Objaverse dataset.

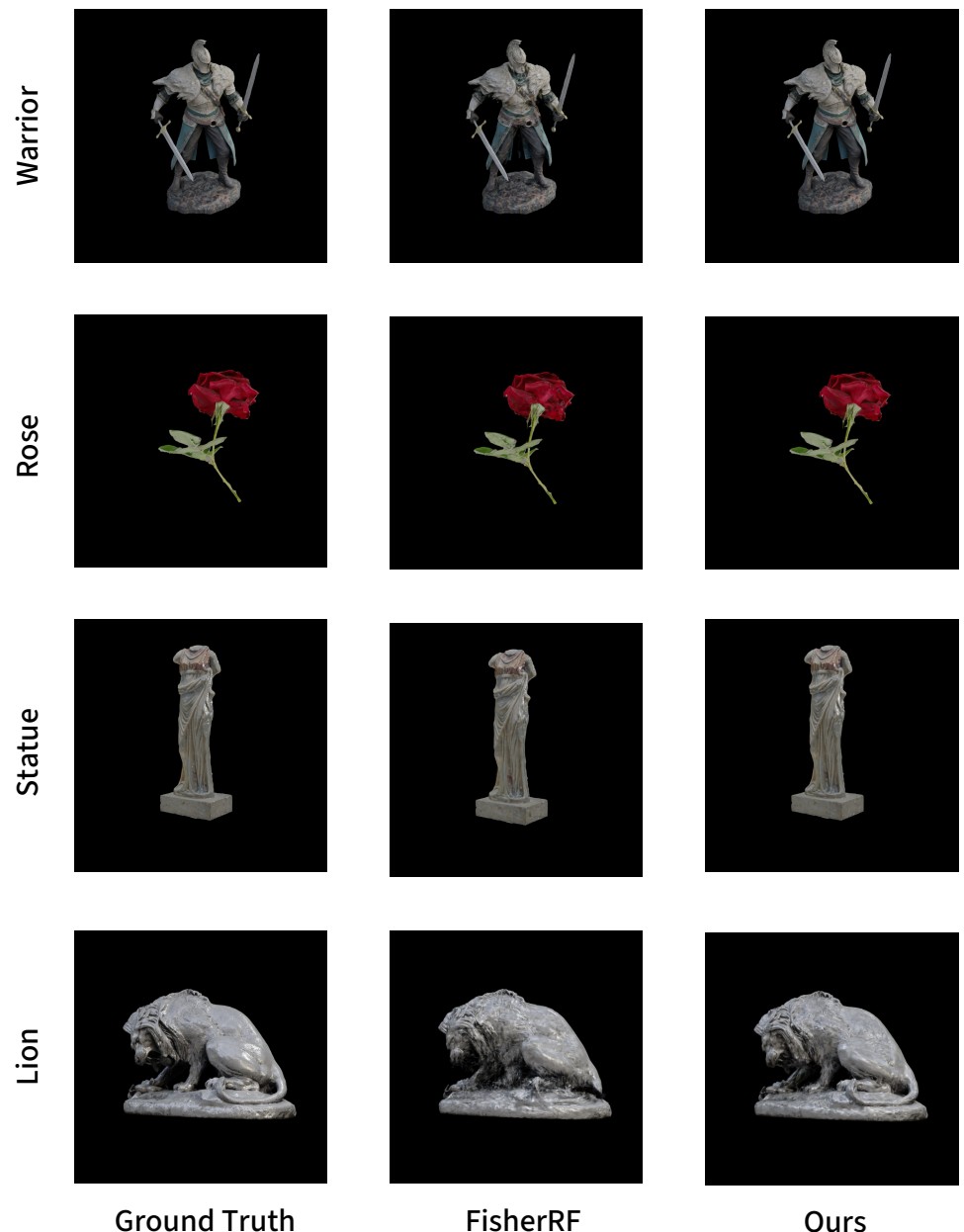

Figure 9: The per-scene qualitative results of rendered images using active reconstruction using 20 total views on the scenes from the Objaverse dataset.

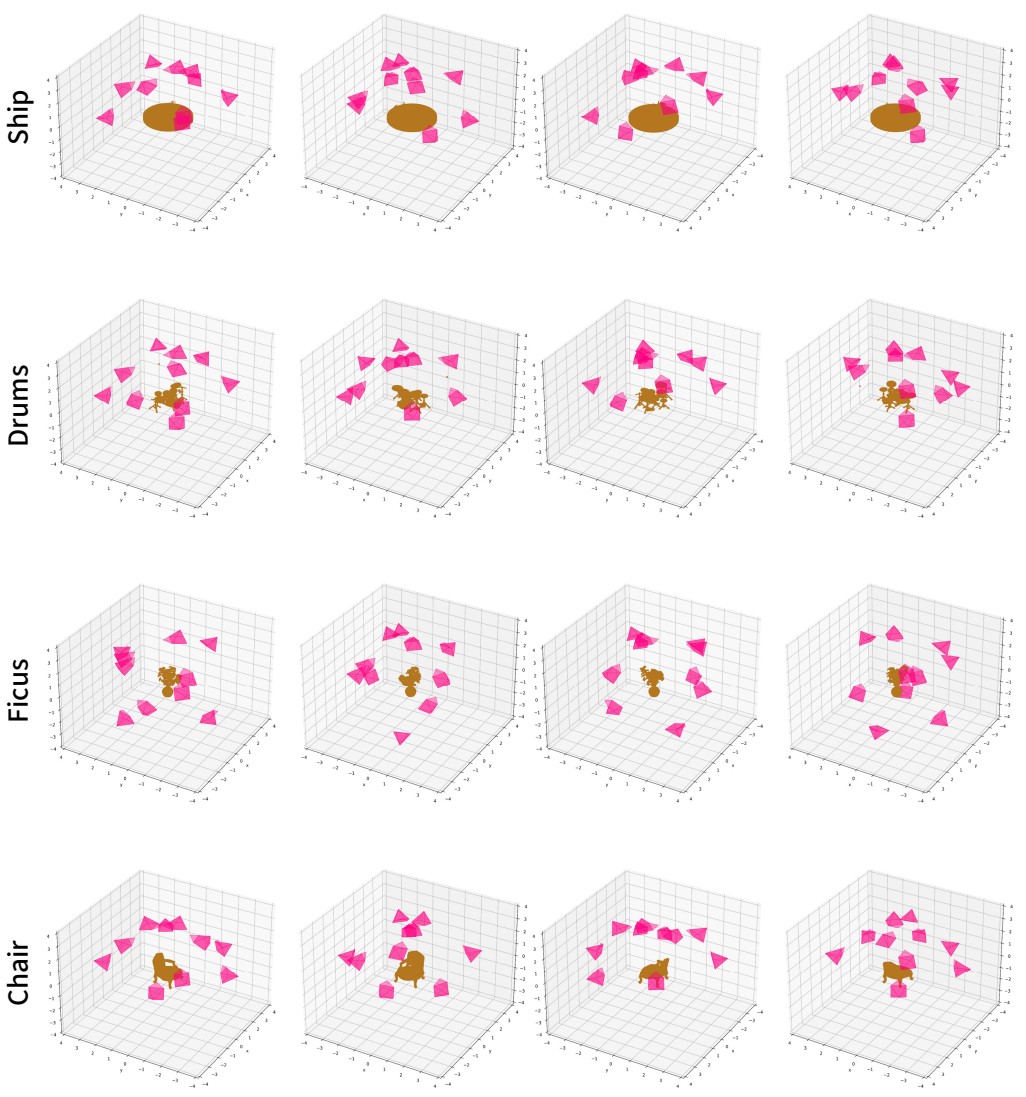

Figure 10: The distribution of all estimated next best views for scenes from the Blender dataset. The total number of views is 10.

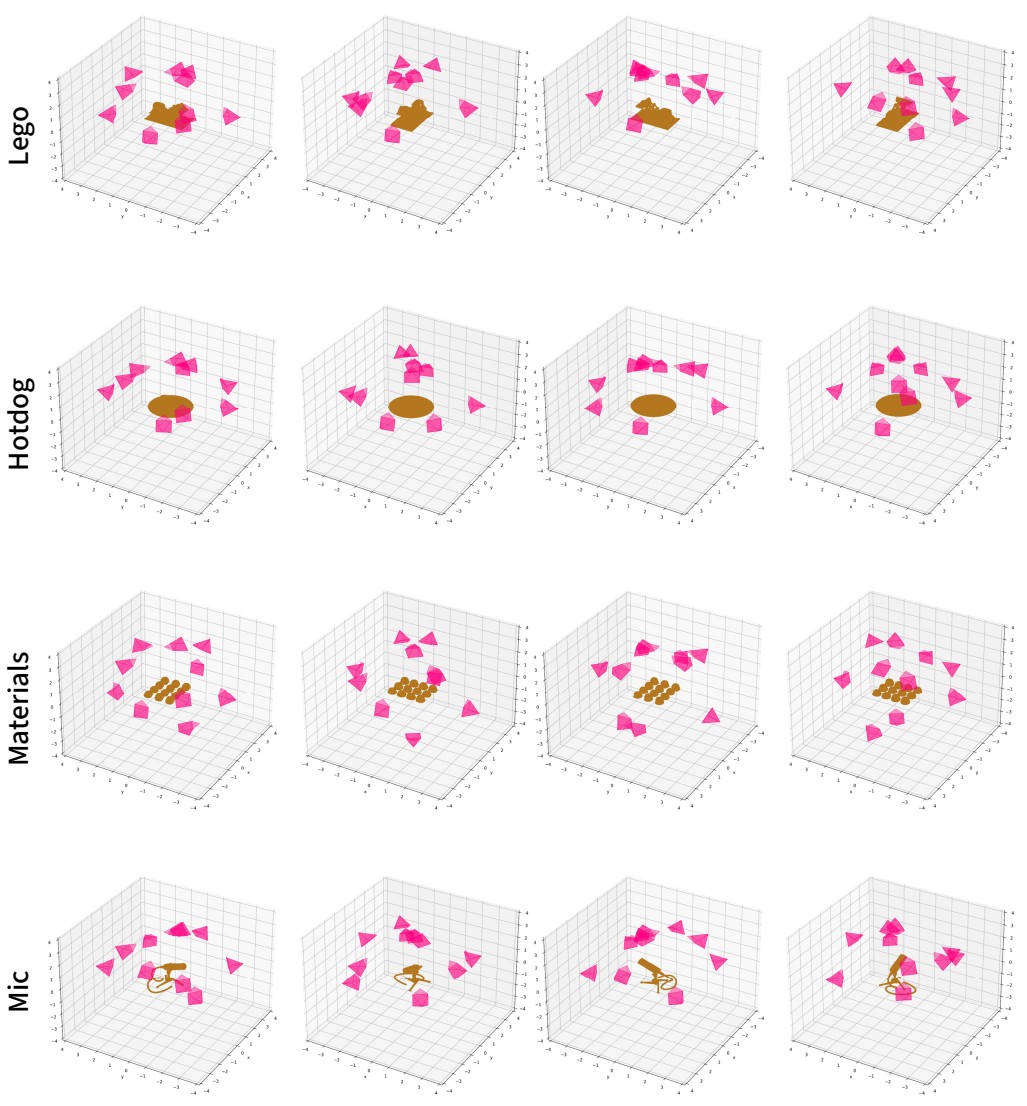

Figure 11: The distribution of all estimated next best views for scenes from the Blender dataset. The total number of views is 10.

