# OpenReview forum: "Active Learning of 3D Gaussian Splatting with Consistent Region Partition and Robust Pose Estimation"
_ICLR.cc/2026/Conference — ICLR 2026 Poster_

### Official Review · Reviewer_PDuY · 2025-10-29

**Soundness:** 3
**Presentation:** 3
**Contribution:** 3
**Rating:** 4
**Confidence:** 4

**Summary:**

This paper proposes an *active learning* framework for 3D Gaussian Splatting (3DGS) that jointly performs image acquisition and model training based on the following key findings.

- Consistent Region Partition: the 3DGS is clustered into local visibility-consistent regions using K-Means on a combined feature of position, color, rotation, and visibility.
- Semantic Feature Variance Score: the region are estimated via variance in Point-MAE semantic features across samples, guiding next-view selection.
- Pose Optimization: a robust photometric pose refinement desired best capture poses and actual capture poses in real-world.

Experiments on NeRF-Synthetic, real-world handheld captures, and Objaverse show quantitative improvement and demonstrate robustness to pose noise.

**Strengths:**

- Idea for online 3DGS training and the capture loop is well motivated and demonstrated via real-world experiment
- Utilizing point-wise metrics for active learning seems novel compared to demonstrated existing works using image-based metrics.
- Comprehensive evaluation on synthetic and real data demonstrates robustness to pose noise.

**Weaknesses:**

1. **Overlap with prior sampling methods.**

    The von Mises-Fisher camera sampling still relies on probabilistic view selection similar to random hemisphere sampling; a clear distinction from ActiveNeRF or FisherRF seems missing.

2. **Heuristic semantic variance metric.**

    The use of dropout variance from Point-MAE features is empirical. An analysis of dropout ratio or random sampling rate effects would be great (Table 4 only partially covers this).

    Moreover, testing other point-cloud encoders[1][2] could clarify whether improvements generalize beyond Point-MAE.

3. **Pose optimization noise propagation.**

    Because $\mathcal{L}_{pose}$ in Eq.(7) uses rendered images for photometric alignment, any render artifact can bias pose updates. An experiment quantifying pose error after refinement would clarify robustness.

4. **Lack of qualitative comparison on NeRF-Synthetic dataset**

    Additonal qualitative comparison on NeRF-Synthetic would strengthen claims of better inconsistent-region handling.

5. **Re-initialization strategy unclear.**

    The active loop re-initializes the 3DGS every 300 steps, but it is unclear how densification and opacity reset interact with semantic variance tracking. Ablation on this would be helpful.

[1] Zhao, Hengshuang, et al. "Point transformer." *Proceedings of the IEEE/CVF international conference on computer vision*. 2021.

[2] Park, Chunghyun, et al. "Fast point transformer." *Proceedings of the IEEE/CVF conference on computer vision and pattern recognition*. 2022.

**Questions:**

Please see the suggestions and comments from the Weaknesses section above.

Additionally:

- In Figure 4, could you explicitly mark the newly added views at each iteration to help visualize coverage progression?

---

> ### Author Response · Authors · 2025-11-25
> **Author response (1/2)**
>
> We thank reviewer PDuY for acknowledging the novelty of our point-wise semantic metric, the motivation of our active-learning design for online 3DGS training, and the comprehensiveness of our evaluation. Below, we respond to the detailed questions and feedback raised by reviewer PDuY.
>
> ### **W1 Viewpoint Diversity and Region Partitioning**
>
> Our method differs fundamentally from prior sampling approaches because the von Mises–Fisher distribution is applied after consistent-region partitioning identifies a specific, concentrated under-reconstructed region. We first partition the point cloud into clusters based on geometry and visibility to ensure each region is within a distinct object part, and thus to further generate viewpoints around different parts. The resulting vMF sampling is therefore directional and region-focused, concentrating poses around the mean viewing direction of the selected region, unlike random hemisphere sampling, which is uniformly distributed and untargeted. This makes our acquisition strategy qualitatively different from both random sampling and methods like ActiveNeRF or FisherRF that rely on global candidate evaluation rather than targeted local refinement.
>
> ### **W2  Heuristic semantic variance metric**
>
> To further validate the robustness of our semantic-variance metric, we conduct two additional ablations on the dropout rate and the point cloud semantic model we used, and report the average novel synthesis performance on testing views after reconstructing 8 scenes in the NeRF-Synthetic dataset using a total of 10 images.
>
>
> 1. Varying the dropout probability $p$ in the Point-MAE encoder:
>
> | Dropout prob. p | PSNR ↑ | SSIM ↑ | LPIPS ↓ |
> |-----------------|--------|--------|---------|
> | 0.5         	| 24.083 | 0.842  | 0.118  |
> | 0.8 (default)   | 25.542 | 0.896  | 0.063  |
> | 0.9         	| 25.508 | 0.891  | 0.066  |
>
> Our default dropout rate is $p=0.8$, and we test with $p=0.5$ and  $p=0.9$ to study the reconstruction performance variation. A lower dropout setting would lead to clearly degraded performance across all metrics, indicating that insufficient dropout reduces semantic feature diversity and weakens the reliability of the variance estimate. In contrast, $p=0.8$ and $p=0.9$ yield stable results with only slight variation. This confirms that the proposed semantic variance metric is robust once the dropout rate is set in a reasonable range, and does not rely on precise hyperparameter tuning.
>
> 2. Replacing Point-MAE with different point-cloud encoders.
>
> To address the reviewer’s concern regarding alternative choices other than the Point-MAE, we evaluated whether the semantic variance metric generalizes to other point-cloud encoders. The public Point Transformer [3] implementation lacks a pretrained checkpoint, and training one from scratch would exceed a reasonable computational budget, so we did not include it in our comparison. We used the pretrained Fast Point Transformer [4] to replace the point feature extractor in our work. The results are shown below:
>
> | Encoder backbone | Number of Parameters (Million) | PSNR ↑ | SSIM ↑ | LPIPS ↓ |
> |-------|----|-----|------|-----|
> | Point-MAE (default) 	| 27.05 | 25.542 | 0.896  | 0.063  |
> | Fast Point Transformer  | 37.86 | 25.487 | 0.892  | 0.069  |
>
> As shown in the Table above, Fast Point Transformer achieves performance that is very close to Point-MAE, which shows that our semantic variance formulation is not tied to a particular feature extractor. Although the two backbones differ significantly in architecture and training objectives, both provide point-level embeddings that lead to similar downstream behavior in our active reconstruction pipeline. This indicates that our method only requires a point-cloud encoder capable of producing meaningful features, and it can be used with a wide range of point-cloud understanding networks.
>
> [3] Zhao, Hengshuang, et al. "Point transformer." CVPR, 2021.
>
> [4] Park, Chunghyun, et al. "Fast point transformer." CVPR, 2022. https://github.com/POSTECH-CVLab/FastPointTransformer

---

> ### Author Response · Authors · 2025-11-25
> **Author response (2/2)**
>
> ### **W3  Pose optimization noise propagation.**
>
> To evaluate the performance of the pose refinement module, we compare our estimated camera poses against an external Structure-from-Motion method on all 4 objects from the AR-based experiment. Specifically, after completing AR-based active capture of the four scenes, we run COLMAP on the full set of acquired images and treat the resulting poses as ground truth. For each image, we measure the angular difference between the two rotations and the normalized translation error in the 1^3 sized world coordinate system.
>
> | Scene   | Rotation Error (deg) | Translation Error|
> |-----|----------|--------|
> | gundam  |  3.8	| 0.014  |
> | phoenix | 10.6	| 0.093  |
> | horse   |  9.7	| 0.041 |
> | teddy   |  4.5	| 0.022  |
> | Average |  7.4	| 0.048   |
>
> As shown in the above Table, the rotation errors vary between different scenes with a maximum error around 10 degree, and the translation error is typically less than 0.05, corresponding to only a small fraction of the scene’s spatial extent. These values indicate that our estimated poses remain consistently close to the COLMAP solution, despite relying on rendered images for photometric alignment. This demonstrates that render artifacts do not significantly bias pose updates, and that the pose-refinement stage is robust and does not accumulate noticeable error in practice.
>
> ### **W4 Lack of qualitative Comparison in Blender**
>
> We appreciate the reviewer’s suggestion. To strengthen the qualitative evaluation, we have updated Figure 6 and Figure 7 in the revised paper to include direct comparisons with FisherRF on the NeRF-Synthetic dataset. The new visualizations clearly show that FisherRF frequently produces noisy reconstruction under both 10 and 20 total view cases, indicating that our semantic variance accurately recognizes the region that needs additional image supervision, thus guiding well the active reconstruction. These results further support the validation of our method.
>
> ### **W5 Unclear Re-initialization strategy**
>
> | Densification Interval | PSNR ↑ | SSIM ↑ | LPIPS ↓ |
> |-----|---|-----|-----|
> | 50| 25.513 | 0.894  | 0.060   |
> | 100 (default)   | 25.542 | 0.896  | 0.063  |
>
> Thank you for raising the concern about the 3DGS density control setting during the view selection stage. We test alternative densification intervals of 50 and 100 steps, and the averaged results of active reconstruction of all 8 scenes from the Blender dataset with a total of 10 images are shown in the above Table. The reconstruction performance changed only marginally. This is because each acquisition cycle uses 300 training steps, which is sufficient for the model to recover the coarse geometry from the currently available views but not long enough for aggressive densification dynamics to materially affect the outcome. For the same reason, opacity reset is not triggered under our 1,000 step interval, so it does not interact with the semantic variance computation. Overall, the densification interval and opacity behavior have little effect on our estimated next view.
>
>
> ### **Q1: In Figure 4, could you mark the newly added views**
>
> We thank the reviewer for the helpful suggestion. We have updated Figure 4 to explicitly highlight the newly added view at each NBV iteration. In the revised figure, the newly selected camera is marked in green, making it easier to visualize how the coverage expands over time. The selected view captures the region of poor reconstruction quality as shown in the second row, indicating that our method provides complete coverage for the object under reconstruction. This update improves the interpretability of the figure and more clearly illustrates the progression of our active view acquisition process.

---

> > ### Comment · Reviewer_PDuY · 2025-11-28
> >
> > I appreciate the authors' detailed response and revision which have addressed my initial concerns.
> > In particular, the responses have shown robustness of the proposed components (W2, W3, W5), clarified the novelty and the concept of the work (W1, Q1), and provided visual comparisons (W4) showing enhanced reconstruction results.
> >
> > I believe the paper's contribution to online active learning of Gaussian primitives is sufficiently practical, and I will raise my rating to 6.
> >
> > However, I still have an additional comment regarding the semantic feature variance. It would further strengthen the paper if the motivation behind using semantic feature variance could be discussed in connection with prior work from related domains, such as [1]. I would appreciate it if the authors could provide such a discussion.
> >
> > [1] Kendall, Alex, and Yarin Gal. "What uncertainties do we need in bayesian deep learning for computer vision?." Advances in neural information processing systems 30 (2017).

---

> > > ### Author Response · Authors · 2025-11-29
> > >
> > > We sincerely thank the reviewer for the detailed review of our work and the willingness to raise the rating. We are encouraged that our responses regarding the robustness of the method, the clarification of novelty, and the additional visual comparisons have successfully addressed your initial concerns.
> > >
> > > Regarding the additional comment on the semantic feature variance: we thank the reviewer for this insightful suggestion. Connecting our method to the Bayesian deep learning framework has significantly strengthened our theoretical justification. As established in prior works, including Kendall & Gal [1], uncertainty in deep learning can be categorized into aleatoric uncertainty (noise inherent in the data) and epistemic uncertainty (uncertainty in the model caused by a lack of knowledge or data). In the context of active learning, the primary goal is to reduce epistemic uncertainty, as this type of uncertainty can be reduced by acquiring more data from the relevant regions.
> > >
> > > Our semantic feature variance metric is designed to estimate precisely the epistemic uncertainty regarding the semantic feature of the 3D scene. By performing multiple stochastic forward passes with Monte Carlo Dropout (MC Dropout) through the Point-MAE encoder, we measure the variance of the resulting features. Our approach builds upon the technique proposed by Kendall & Gal [1], where they also utilize MC Dropout to approximate the posterior distribution of model weights and estimate uncertainty via the variance of the model's predictive distribution across multiple stochastic forward passes. However, we adapt their approach by applying stochastic dropout within the Transformer blocks of our Point-MAE encoder during inference. This design suits our specific problem domain: instead of measuring variance in a final output, we compute the variance in the semantic feature space. A high variance in this feature embedding indicates that the model is uncertain about that region, likely due to poor geometric reconstruction or insufficient coverage, and thus identifies it as a region of high epistemic uncertainty. By targeting these high-variance regions for the next view acquisition, our method adheres to the principled active learning strategy of maximizing information gain by reducing model uncertainty.
> > >
> > > We have updated the submission by including the above discussion and the reference to [1] in the justification section in the Appendix Section A.5.2.
> > >
> > > [1] Kendall, Alex, and Yarin Gal. "What uncertainties do we need in bayesian deep learning for computer vision?." Advances in neural information processing systems 30 (2017).

---

### Official Review · Reviewer_dCpe · 2025-10-31

**Soundness:** 3
**Presentation:** 3
**Contribution:** 4
**Rating:** 6
**Confidence:** 4

**Summary:**

This paper tackles active view selection for 3D scene reconstruction with 3DGS, addressing how current radiance field workflows capture many images empirically without guidance, causing redundant or insufficient coverage. The authors propose a bottom-up active learning algorithm that partitions the scene's Gaussian point cloud into spatially coherent regions based on attributes and visibility, then estimates semantic feature variance per Gaussian to identify the most informative regions to explore next. This semantic variance acts as a high-level uncertainty measure revealing model incompleteness. The method also incorporates robust pose optimization to refine noisy camera poses from sources like SLAM or AR sensors. Extensive experiments on synthetic and real-world datasets show the approach achieves faster, higher-quality reconstruction than existing methods, contributing a novel region-based selection framework, a semantic-level uncertainty metric for 3DGS, and improved pose noise robustness that collectively enhance radiance field acquisition efficiency.

**Strengths:**

- A hard-but-important problem tackled with a clean, region-wise idea: cluster Gaussians by visibility to get consistent parts, then use semantic feature variance to pinpoint what’s under-reconstructed—more direct than image-space heuristics and easy to plug into 3DGS.
- The pose-refinement loop makes handheld/AR capture practical rather than brittle. Results on NeRF-Synthetic are strong in low-view regimes, and ablations show each module matters.
- Reproducibility is treated seriously: the authors provide NeRF-Synthetic results and partition meshes (colored by region) so readers can inspect the visibility-based separation, and they commit to open-sourcing the full code, which is great for the community.

**Weaknesses:**

- Although plausible, the semantic-feature variance remains a heuristic; it’s not rigorously tied to reconstruction error nor carefully pitted against simpler 3DGS-native cues (*e.g.*, RGB/opacity variance or Fisher information).
- Region partitioning introduces knobs (cluster count/thresholds) whose stability is only lightly explored; a small robustness sweep and an adaptive stopping rule would usefully de-risk it.
- The pipeline is heavy—visibility sampling, multiple feature passes, clustering, then pose refinement—yet there’s no wall-clock or memory breakdown per NBV iteration, so speed/latency remains unclear for both human-in-the-loop capture and embodied agents (*e.g.*, AR round-trip, pose-refinement iterations, and how N/T/K trade time vs. quality).
- The design is tailored to 3DGS; portability to NeRF/voxel backbones without explicit splats is non-obvious.
- Pose refinement’s convergence basin and failure modes (large drift, low-texture regions) are not quantified, leaving robustness bounds somewhat ambiguous.
- To ensure a fair evaluation of performance, the results for the 12 objects from the Objaverse dataset should be reported. It would also be helpful for the authors to provide a video showing the method's performance across 3 to 20 views, along with comparisons to other methods.
- Minor: Line 75 "e.g." should be italicized, meaning it should appear as *e.g.* . Some descriptions are ambiguous, making them hard to understand. For instance, Line 230-234. The cite of Alpha shape should be placed on Line 267 rather than Line 292.

**Questions:**

1. Figure 3 shows the local consistent region determination, and I wonder about the input view distribution of this demo.
2. I have noticed that “The image acquisition is performed every 300 steps, after which the model is reinitialized as random points.” Why is this design? Why not continue training?
3. After collecting all the images, the model undergoes further training for 10,000 steps. Do other models follow the same process, considering that 3DGS is sensitive to the number of training iterations?
4. What specific “semantic-level features” are used to compute the Gaussian feature variance, and how robust are they across different scenes?
5. Additionally, the initialization pose is an important consideration. Since this paper focuses on practical applications, the initial pose should be addressed. In sparse settings, where acquiring an accurate pose is challenging, many methods utilize neural matching approaches like VGGT, MASt3R, or DUSt3R. It would be interesting to explore whether these methods can be integrated into the pipeline, similar to GaussianObject. These methods should also be cited.
6. The authors claim that "The mesh with after consistent region partition can be found in folder 'partition_results'. ", but none of this folder is provided.
7. How is the consistent region partition determined in practice, and how sensitive is the method to this partitioning?

---

> ### Author Response · Authors · 2025-11-25
> **Author response (1/4)**
>
> We thank reviewer dCpe for recognizing the importance of our region-based active view selection framework and for noting the practical relevance of our pose-refinement module for handheld/AR capture scenarios. We appreciate the reviewer’s positive comments on our clean methodology, strong low-view performance, and commitment to reproducibility.
> Below, we address the specific concerns and suggestions.
>
> ### **W1 Semantic Feature Variance vs 3DGS-Native Cues**
>
> We appreciate the reviewer's concern regarding the rigorous theoretical grounding of the semantic-feature variance. We maintain that the semantic feature variance is a motivated 3D-native metric that serves as a proxy for per-point reconstruction error, particularly concerning perceptual quality and geometric uncertainty. We mentioned in Section A.5.2 that the variance is a direct quantification of epistemic uncertainty derived from multiple forward passes through a Point-MAE encoder pre-trained on ShapeNet, which injects extensive prior knowledge of 3D object geometry. This is a standard and robust way to quantify model uncertainty, where high variance directly indicates a poor or unstable feature representation. This metric overcomes the limitations of 3DGS-native cues such as RGB/opacity variance or Fisher Information, which primarily reflect low-level and local texture or lighting variations, and cannot capture the high-level semantic quality that is critical for perceptual quality. We have updated the justification in  Section A.5.2 to strengthen the motivation of semantic feature variance.
>
> ### **W2 Cluster Number Ablation**
>
> In Appendix Table 3, we present an ablation study on the cluster number, demonstrating that $K=5$ is the optimal balance for our approach. Higher values increase sensitivity to noise while lower values yield coarse partitions. While we use a fixed, empirically validated $K=5$ in our experiments, we recognize that the partitioning is inherently compatible with an adaptive stopping rule for future improvement; specifically, we can integrate an approach where clustering terminates when the minimum per-cluster semantic variance falls below a threshold, offering an adaptive $K$ value that adjusts to the scene complexity.
>
> ### **W3 Runtime and Memory Breakdown**
>
> To make the speed and latency characteristics clearer, especially for human-in-the-loop capture, we now provide a detailed runtime breakdown across all baselines:
>
> | Method | Time Interval per View Selection Step | Total Reconstruction Time    	|
> |----|----|----|
> | ActiveNeRF | ~5 min | ~2 hours   |
> | 3DGS + Furthest/Random | N/A | 397 seconds 	|
> | FisherRF  | 17 seconds | 754 seconds 	|
> | **Ours** | **24 seconds** | **474 seconds**	|
>
> While the time interval per view selection step of our method is moderately higher than FisherRF, this difference arises mainly from the pose-refinement stage, which is necessary for handling noisy camera poses. This setting is not considered by FisherRF or ActiveNeRF. Importantly, this additional refinement does not translate to higher overall latency. Despite performing more operations per step, our approach achieves a lower total reconstruction time.
>
> This improvement stems from the fact that the views selected by our semantic-driven NBV strategy are substantially more informative than those of prior methods. As a result, our model requires significantly fewer total training iterations to reach high-quality reconstruction. Concretely, our final model converges within 10K training steps, whereas FisherRF typically needs 30K steps to reach comparable performance. The reduction in global training effort therefore compensates for, and ultimately outweighs, the per-step overhead.
>
> Regarding the reviewer’s question about how the variables N, T, and K influence runtime, we note that the number of points in each consistent region (N) and the number of clusters (K) contribute minimally to computational cost, since they affect only the lightweight visibility matrix estimation and clustering stages. The number of semantic feature samples (T) has an impact on runtime because it increases the workload of the Point-MAE inference. However, all samples are processed in a single batched forward pass, allowing efficient parallel execution and preventing runtime from scaling linearly with T.
>
> For memory usage, our training stage retains the same footprint as standard 3DGS. The only additional memory arises during NBV estimation, where semantic features must be extracted. This step introduces approximately 9534 MB of extra peak GPU memory; however, the memory would be freed after the semantic feature inference stage, and could be further optimized with engineering approaches such as quantization or mixed-precision inference. We hope these clarifications can address the reviewer’s concerns about pipeline runtime and memory costs.

---

> ### Author Response · Authors · 2025-11-25
> **Author response (2/4)**
>
> ### **W4 Generalization Beyond 3DGS**
>
> Actually, our method can generalize to NeRF and voxel-based methods. The consistent region partition and semantic feature variance in our method are model-agnostic and can be effectively ported to implicit methods like NeRF or voxel-based backbones. This is achieved by first extracting a sparse point cloud from the dense reconstruction of the NeRF or voxel method, using density thresholds or marching cubes. Once this point cloud is obtained, our method processes the points using the Point-MAE encoder to compute semantic features, followed by region partitioning and view selection. These steps only require geometry and visibility information, both of which are readily accessible or derivable from the outputs of implicit radiance field representations.
>
> ### **W5 Pose Optimization Convergence**
>
> We thank the reviewer for highlighting the need to characterize the robustness of our pose refinement module more explicitly. To address this, we conducted an additional controlled experiment on the Lego scene from the NeRF-Synthetic dataset, where we injected synthetic pose noise of increasing magnitude. In this setup, rotational noise was sampled from a zero-mean Gaussian with progressively larger standard deviations, and translation noise was added proportionally to the scene scale. This controlled setting allows us to directly examine how refinement behaves under different levels of drift.
>
> | Rotation Noise std (degree) | Translation Noise std | Pose Optimization |  PSNR  | SSIM  | LPIPS |
> |-------|---|----|---|---|--------|
> | 0.0   | 0.00  | -    | 22.91  | 0.868 | 0.119 |
> | 10   	| 0.01  | **On** 	| **22.27**  | **0.857** | **0.125** |
> | 10   	| 0.01  | Off   	| 21.18  | 0.836 | 0.142 |
> | 20    | 0.03  | **On**     	| **22.15**  | **0.851** | **0.129** |
> | 20    | 0.03  | Off     | 20.47  | 0.822 | 0.154 |
> | 30    | 0.05  | **On**  	| **21.19**  | **0.833** | **0.145** |
> | 30    | 0.05  | Off     | 19.38  | 0.796 | 0.171 |
>
> As shown in the above table, when the pose perturbation is mild or moderate, the refinement module is able to recover reconstruction quality very close to the noise-free baseline. For example, under moderate noise, the variant with pose refinement maintains a PSNR that remains within a small margin of the baseline, while the training without refinement drops substantially. This demonstrates that our optimization reliably compensates for the level of drift typically observed in object-level capture. As the injected noise becomes large, the recovered PSNR decreases, and the gap between the refined and unrefined variants widens. This behavior is consistent with the expected limitations of photometric pose optimization under severe misalignment.
>
> Overall, the results illustrate the convergence basin of pose refinement. Our method can effectively correct small to moderate drift, maintaining reconstruction performance close to the original baseline, while extreme cases might lead to degradation in the reconstruction results.
>
> Regarding large drift, we agree that our method may not correct extreme cases, such as when the initial pose is on the opposite side of the object; in such scenarios, the photometric objective becomes too ambiguous for optimization to converge to the correct basin. However, in our object-level active capture setting, the camera viewpoints typically remain within a limited angular neighborhood.
>
> For low-texture regions, we observe that object shape, together with some regional features, can also provide strong cues even when there exist some local low-texture regions. For example, on objects like the Hotdog, which contains some low-texture surfaces, our refinement can still recover the correct pose due to the global consistency enforced by the 3D Gaussian structure.

---

> ### Author Response · Authors · 2025-11-25
> **Author response (3/4)**
>
> ### **W6 Objaverse Results and Videos**
>
> We apologize for the confusion in the earlier version regarding the number of objects used from the Objaverse dataset. To clarify, apart from the NeRF-Synthetic benchmark, our experiments in the Appendix include results for 12 objects in total, comprising 8 Objaverse objects and 4 AR-based active-reconstruction objects. The corresponding qualitative and quantitative results are presented in Figure 8, Figure 9, and Table 10. We also include comprehensive videos in the updated supplementary files, covering both the testing comparisons with the baseline method and the full training progress visualization of our method. Each frame in the video is recorded every 50 steps during the whole pipeline, firstly the Next-Best-View adding stage and then the final training stage. Therefore, the video shows the training results as new views are gradually added, from 3 initial views to 20 total views. The training videos illustrate how the reconstruction gradually improves in both quality and completeness as new views are added. The testing video shows that our method consistently achieves better visual quality than the baseline in the testing views, demonstrating its superiority. Videos are provided for all 8 objects from the Objaverse dataset.
>
> ### **W7 Writing Style and Citations**
>
> We thank the reviewer for the careful review of our manuscript. We have addressed all raised points by italicizing *e.g.* on Line 75, and moved the citation for Alpha shape to the suggested position on Line 267. We have rewritten the whole paragraph of the description of our robust pose estimation in the original Lines 230-234 to enhance clarity and precision.
>
> ### **Q1 Distribution of Input Views and Region Partitioning**
>
> In Figure 3, to visualize a complete point cloud model, we use a point cloud of the drum scene from the NeRF-Synthetic dataset, reconstructed from 20 views, where the object geometry is clearly reconstructed, and the sparse view setting aligns with the case of our algorithm. Additionally, if the input view distribution refers to the input view used for calculating the local consistent regions, then we sampled 100 dense views from the hemisphere around the object to determine the visibility matrix, which generates a stable consistent region partitioning result.
>
> ### **Q2 Justification for Reinitialization**
>
> Actually, we initially attempted to continue training the model using the newly added views. However, this turns out to be ineffective because the initial dataset was sparse, causing the early 3D Gaussian Splatting model optimization to introduce incorrectly densified Gaussians, which led to the model overfitting to the initial input views. Consequently, when new, accurate views were added, the model was unable to optimize or prune the densified wrong points effectively. Therefore, we use the reinitialization strategy to reset the optimization states and break this overfitting situation and restore the model training status, ensuring the model can efficiently learn the new information acquired from the added view.
>
> ### **Q3 Total Number Training Iterations**
>
> Actually, other methods like ActiveNeRF and FisherRF typically use a total of 30,000 training steps, whereas our method requires only 10,000 additional steps, resulting in a significantly smaller overall training budget. This reduction in training iterations leads to faster reconstruction, while our method still achieves higher visual quality in the synthesized novel views. These results demonstrate that our method selects a more informative set of views, enabling higher-quality reconstructions with fewer total training steps.

---

> ### Author Response · Authors · 2025-11-25
> **Author response (4/4)**
>
> ### **Q4 Definition and Use of Semantic Features**
>
> We now clarify in Section A.2.2 that the semantic-level features correspond to Point-MAE transformer encoder embeddings. These embeddings capture higher-level semantic features that cannot be represented by raw RGB or opacity statistics. Because Point-MAE is pretrained in a self-supervised manner on large and diverse point-cloud datasets, the resulting features are inherently robust across different object classes and shapes.
> In Table 6, Table 7 and Table 10 of the Appendix, the per-scene results on the NeRF-Synthetic and Objaverse dataset further demonstrate that this informativeness metric consistently achieves higher performance across all different object classes, confirming its robustness and generality. These findings show that our proposed semantic features provide a stable signal for identifying under-reconstructed regions, directly contributing to the overall performance gains of our method.
>
> ### **Q5 Pose Initialization and Integration with Learned Estimators**
>
> We appreciate the suggestion for improving pose estimation using the latest methods. While our current system operates under the proposed pose optimization method, we agree that integrating learning-based methods is a valuable direction to explore, especially for real-world sparse-view cases. The methods referred to by the reviewer, such as VGGT, MASt3R, DUSt3R and GaussianObject, represent tools and frameworks for obtaining accurate initial poses from sparse inputs. We have enhanced the manuscript by adding a discussion in Section A.11 that outlines how these techniques can be integrated into our pipeline. We think this integration would be a fundamental enhancement to the utility of our active learning framework.
>
> ### **Q6 Missing Supplementary Files**
> Thank you for pointing out this omission. The meshes have now been uploaded to the folder 'partition_results' in the supplementary files of the revised submission. These meshes show the intermediate results of our consistent-region partitioning in Figure 4 and Figure 5, demonstrating that our method produces reasonable, stable partitions even when applied to noisy meshes generated by 3DGS reconstruction.
>
> ### **Q7 Partitioning Variability and Result Stability**
>
> The consistent-region partition is determined through the two-stage procedure described in Section 3.2.1 of the main paper. Specifically, we first construct per-Gaussian features that combine position, color, rotation, and visibility, and then apply K-Means clustering to group Gaussians into visibility-consistent regions. This allows the algorithm to isolate coherent surface parts that share similar geometric and visibility characteristics.
> Although this partitioning process involves stochastic components, such as feature sampling and K-Means initialization, we evaluated its stability by running all experiments with three different random seeds. The reconstruction results are highly consistent across three independent runs, with PSNR variance remaining below 0.1 dB. This demonstrates that the partitioning strategy is robust and does not introduce sensitivity into the final reconstruction quality.

---

### Official Review · Reviewer_fPM1 · 2025-11-01

**Soundness:** 3
**Presentation:** 3
**Contribution:** 2
**Rating:** 6
**Confidence:** 3

**Summary:**

This paper presents a novel active learning framework for 3D Gaussian Splatting (3DGS) that guides view acquisition during reconstruction. The proposed method introduces a bottom-up strategy that partitions the model into locally consistent regions based on Gaussian attributes and visibility features, identifies noisy or under-reconstructed areas via a semantic feature variance metric, and generates the next best views accordingly. To further enhance robustness in real-world capture scenarios, the authors incorporate a pose optimization module that refines noisy camera poses using reprojection constraints. Extensive experiments on both synthetic and real-world datasets demonstrate that the approach achieves state-of-the-art reconstruction quality under both accurate and noisy pose settings, validated through comprehensive quantitative comparisons and ablation studies.

**Strengths:**

Quality and Clarity: The paper is clearly written and well-structured, with a logical flow from motivation to methodology and experiments. The technical explanations are concise yet sufficiently detailed, making the method easy to follow. Figures effectively illustrate key components such as consistent region partitioning and pose optimization, which enhances overall readability and technical transparency.

Significance: The experimental results demonstrate state-of-the-art performance across both synthetic and real-world datasets, validating the robustness and effectiveness of the proposed method. The paper’s combination of semantic feature variance for uncertainty estimation and robust pose refinement addresses important limitations of prior active reconstruction approaches, showing strong potential impact for practical 3DGS-based active learning systems in robotics and AR/VR applications.

**Weaknesses:**

Incomplete coverage of recent literature: While the paper provides a solid overview of classical active reconstruction and next-best-view methods, it omits several recent and highly relevant works such as Naruto: Neural Active Reconstruction from Uncertain Target Observations, ActiveGAMER: Active Gaussian Mapping through Efficient Rendering, and Understanding while Exploring: Semantics-driven Active Mapping. These methods also explore uncertainty-driven or semantic-guided active mapping. A comparative discussion or experimental inclusion of these approaches would better contextualize the paper’s contributions and clarify its novelty.

Lack of empirical validation for efficiency claims: The paper repeatedly highlights the proposed bottom-up strategy’s efficiency in generating informative views without candidate sampling, yet no quantitative results support this claim. Metrics such as runtime per iteration, total reconstruction time, or convergence rate compared with baselines like FisherRF or ActiveNeRF would be valuable for substantiating the claimed computational advantage.

Absence of discussion on limitations and failure cases: The paper does not provide any analysis of scenarios where the method might fail or underperform, such as highly transparent or reflective surfaces, cluttered environments, or extreme pose noise. A brief reflection on these limitations—along with potential directions for improvement—would enhance the paper’s credibility and balance.

**Questions:**

1. Appendix A.2.2 says K-Means uses “number of clusters equals 10,” yet Table 3 states “5 (Default)” and also reports results for 3/5/7 clusters. Which value was used for the main results, and how sensitive are gains to this choice?

2. You define \hat{x} = x_j/J as “the center of points,” which seems incorrect dimensionally—should this be \hat{x} = $\frac{1}{J}\sum_{j=1}^{J} x_j$?

3. You “first capture three initial images uniformly.” Do all baselines start from exactly the same three initial views and total budget (including those three)

---

> ### Author Response · Authors · 2025-11-25
> **Author response (1/2)**
>
> We thank reviewer fPM1 for their thoughtful assessment of our work, for highlighting the clarity of our presentation, and for acknowledging the strength of our experimental results across both synthetic and real-world datasets. We appreciate the reviewer’s positive rating of our work. Below, we provide our responses to each point raised.
>
> ### **W1 Discussion on Scene-Level Active Reconstruction Literature**
>
> We thank the reviewer for pointing out recent work such as Naruto, ActiveGAMER, and Understanding While Exploring, which indeed push the frontier of uncertainty-aware active mapping. However, these works primarily address scene-level active reconstruction, often relying on SLAM-based pose tracking. In contrast, our work focuses on object-centric active view selection under the constraints of handheld real-world capture, where poses are noisy and memory is limited, and reconstruction must be rapid and view-efficient. Unlike scene-level settings, which require 3DGS SLAM as the reconstruction backbone, our system uniquely integrates pose refinement, particularly for handheld scenarios where accurate camera tracking is unavailable. Specifically, our designed region-based informativeness planning is tailored to object-level geometry rather than globally navigable scenes. To clarify this distinction, we have added a discussion in Section 2.2, reviewing the key contributions of the aforementioned scene-level methods. We hope this clarification could make our scope and novelty in the object-level active view selection clearer.
>
> ### **W2 Validation for Efficiency**
>
> We thank the reviewer for highlighting the need for clearer quantitative evidence supporting our efficiency claims. While Section A.6 of the original submission reported runtime statistics, we have now added a comprehensive comparison against all major baselines, including ActiveNeRF and FisherRF, evaluated on the scenes from the Blender dataset using a single NVIDIA RTX 4090 GPU.
>
> | Method | Time per View Selection Step | Total Reconstruction Time |
> |----|----|----|
> | ActiveNeRF | ~5 min | ~2 hours   |
> | FisherRF  | 17 seconds | 754 seconds 	|
> | **Ours** | **24 seconds** | **474 seconds**	|
>
> As shown above, our method achieves a notably shorter total reconstruction time than previous approaches while simultaneously producing higher reconstruction quality. Although our view selection interval is slightly longer than FisherRF, this difference mainly stems from the additional pose optimization iterations required to handle noisy-pose scenarios, which FisherRF does not consider.
>
> Importantly, our method still achieves a significant overall speedup. This is because the views selected by our semantic guided strategy are more informative, allowing the model to converge with far fewer total training steps. In practice, our method completes reconstruction with 10K final training steps, whereas FisherRF requires 30K steps to reach comparable performance. As a result, the end-to-end reconstruction time of our approach is substantially lower, validating the efficiency benefits claimed in the paper.

---

> ### Author Response · Authors · 2025-11-25
> **Author response (2/2)**
>
> ### **W3 Robustness in Challenging Conditions**
>
> To complete the discussion of our method, we have added a detailed reflection on the limitations and potential future works in Section A.11 of the revised paper. We also summarize key scenarios where our method may underperform and how they could be addressed as follows:
>
> 1. **Transparent Objects**: Our method focuses on the active reconstruction component and builds upon the naïve 3DGS backbone, which might underperform under transparent and reflective materials. However, later 3DGS frameworks, such as GS-IR [1], extend 3DGS to better handle transparent and specular objects through inverse rendering. Replacing our current backbone with such enhanced variants could generate valid point clouds for calculating the semantic variance score in our method, and enable our view planning framework to operate effectively across a wider range of material properties.
>
> 2. **Cluttered Environments**: Our current system targets single-object scenarios. For cluttered scenes with multiple objects, we can also apply clustering to separate objects; however, the semantic feature extraction network assumes the input point cloud contains one object. Extending our framework to multi-object or scene-level reconstruction remains a substantial revision of the pipeline in the future.
>
> 3. **Extreme Pose Noise**: We explicitly address noisy pose with our pose estimation framework and significantly reduce pose error, as shown qualitatively in Figure 5 and quantitatively in Table 8 and Table 9. However, the optimization based pose refinement cannot deal with extreme cases, such as poses from the other side of the object. Further integrating our method with learning-based methods such as VGGT [2] might be a potential path for solving this problem.
>
> [1] Liang, Zhihao, et al. "Gs-ir: 3d gaussian splatting for inverse rendering." CVPR, 2024.
>
> [2] Wang, Jianyuan, et al. "Vggt: Visual geometry grounded transformer." CVPR, 2025.
>
>
> ### **Q1 Clustering and Semantic Feature Sampling Hyperparameters**
>
> Sorry for the confusion on the choice of hyperparameters. By default, we run the clustering algorithm with K = 5 clusters and sample 10 semantic features to calculate their variance. These values were empirically selected to balance informativeness estimation with runtime, and the ablations on these choices are presented in Table 3 and Table 4 in the Appendix. We have revised the above details in Appendix A.2.1.
>
> ### **Q2 Wording and Clarification**
>
> Thank you for pointing this out. In the revised version, we have corrected this by defining the center as the mean of the points in the updated version of the paper.
>
> ### **Q3 Initialization Pose and Fairness**
>
> Thank you for raising this point. The baselines do not all start from identical initial poses. For example, FisherRF begins with four uniform initial frames, ActiveNeRF uses 2/4 random frames for the 10/20 total view case, while our method initializes from three images uniformly distributed around the object. These initial poses are randomly sampled at the beginning of each run. However, the total view budget, including initial views, is kept consistent across all methods to ensure comparability. Furthermore, we average our results over three runs to account for randomness. Given the matched view budget and averaging evaluation protocol, we believe this setup provides a fair and reproducible comparison.

---

### Author Response · Authors · 2025-11-25
**Summary of Rebuttal and Discussion**

We would like to thank the Area Chair and all reviewers for their effort in handling our submission. Below, we summarize 1. the pre‑rebuttal consensus, 2. the summary of our responses and paper revisions to address the concerns raised by reviewers, and 3. the reviewers’ score updates.

## **Pre-Rebuttal Consensus**

Before the rebuttal, reviewers showed a clear, shared view of the strengths of our work in clarity in writing and logic, novelty of the idea, contribution and  reproducibility of the experiments, and practical impact.
Reviewer fPM1 wrote that "The paper is clearly written and well-structured, ..., from motivation to methodology and experiments".
Reviewer dCpe called our approach "A hard-but-important problem tackled with a clean, region-wise idea, ..., makes handheld/AR capture practical."
Reviewer PDuY highlighted that "Idea ... is well motivated and demonstrated via real-world experiment. ... Comprehensive evaluation on synthetic and real data demonstrates robustness to pose noise. ... Reproducibility is treated seriously."

## **Summary of Response and Paper Revisions**

In our rebuttal, we made several experiments and revisions to address specific concerns raised by each reviewer.

1. **Runtime, latency, and memory analysis (fPM1, dCpe).**
We introduced an efficiency comparison against ActiveNeRF and FisherRF on Blender scenes with a single RTX 4090, reporting time per view-selection step and total reconstruction time. Although our per-step latency is slightly higher due to pose refinement, our semantic-guided NBV converges in lower overall runtime; we also broke down how hyperparameters affect cost and showed that semantic-feature extraction is efficiently implemented with an acceptable memory footprint.

2. **Semantic-feature variance justification and ablations (dCpe, PDuY).**
We linked our metric to epistemic uncertainty estimation, explicitly explaining why high semantic variance indicates model incompleteness more directly than RGB/opacity variance or Fisher information. In addition, we added ablations over dropout probability and Point Transformer encoder, showing that our performance gains are robust across reasonable hyperparameters and backbones.

3. **Region partitioning, viewpoint diversity, and hyperparameters (dCpe, PDuY).**
We extended the ablation on cluster number $K$, demonstrating that our default choice is a stable trade-off, and discussed how the partitioning can be made adaptive in future work. We also clarified that von Mises–Fisher sampling improves random hemisphere sampling by making view selection directional and region-focused, and we added partition result meshes in the supplementary files.

4. **Pose robustness, convergence, and evaluation details (dCpe, PDuY, fPM1).**
We added controlled experiments on NeRF-Synthetic with increasing synthetic pose noise, plus post-hoc pose error evaluation with COLMAP estimated poses, to characterize the convergence basin and limitations of our pose refinement. We further clarified evaluation protocol details regarding initial pose selection and total view budget to ensure a fair comparison across baselines.

5. **Limitations, qualitative evaluation, and implementation details (fPM1, dCpe, PDuY).**
We added a limitations and future-work section discussing challenging scenarios such as transparent/reflective objects, cluttered scenes, and extreme pose drift (including potential integration with VGGT). Finally, we expanded Objaverse and NeRF-Synthetic qualitative results and videos from 3–20 views, marked newly added views in figures, clarified the densification/reinitialization schedule and its interaction with semantic variance, and improved the presentation issues noted by the reviewers.

6. **Positioning w.r.t. recent scene-level work (fPM1).**
We added a dedicated discussion in Section 2.2 contrasting our object-centric, handheld-focused setting with scene-level active mapping methods such as Naruto. This clarifies that our goal is rapid object-level capture with pose refinement, rather than long-horizon SLAM-style exploration.

## **Score Updates from Reviewers**

While Reviewer fPM1 and dCpe retained a positive rating of 6, Reviewer PDuY states the willingness to raise the rating to 6 in the comment. Therefore, all reviewers now hold a positive rating of 6.

We hope this summary helps conclude the review, discussion and revision of Submission16583.

Best regards,

Authors of Submission16583

---

### Meta-Review · Area_Chair_JTsW · 2026-01-07

**Summary:**

This paper proposes an active learning / next-best-view (NBV) framework for 3D Gaussian Splatting (3DGS) aimed at reducing redundant image capture and improving reconstruction quality under limited views and noisy camera poses.

Reviewers agreed the paper is clearly written and the problem is practical/important; concerns centered on contextualization to recent active mapping literature, substantiating efficiency/latency/memory claims, and validating the heuristic design choices (semantic variance, clustering knobs, pose convergence bounds, and protocol fairness).

**Reviewer Concerns:**

Concerns substantially addressed in rebuttal:

**Reviewer Scores:**

fPM1: 6 (unchanged)

dCpe: 6 (unchanged)

PDuY: updated from 4 → 6 (likely, explicitly stated in the reviewer’s follow-up comment)

---

### Decision · Program_Chairs · 2026-01-26

Accept (Poster)